# NECromancer: Breathing Life into Skeletons via BVH Animation

## Abstract

Motion tokenization is fundamental to the development of generalizable motion models, yet existing approaches remain restricted to species-specific skeletons, such as humans, thereby limiting their applicability across diverse morphologies. We present **NECromancer (NEC)**, a universal motion tokenizer designed to operate on arbitrary BVH skeletons. NEC is built upon three core components: (1) an **O**ntology-a**W**are Skeletal Graph Enc**O**der **(OwO)**, which leverages graph neural networks to encode structural priors extracted from BVH files—including joint-name semantics, rest-pose offsets, and skeletal topology—into robust skeletal embeddings; (2) a **T**opology-**A**gnostic **T**okenizer **(TAT)**, which compresses motion sequences into a universal, topology–invariant latent representation, thereby decoupling motion dynamics from morphology; and (3) the **U**nified BVH **U**niverse **(UvU)**, a large-scale dataset that consolidates BVH motions across heterogeneous skeletons (humans, quadrupeds, and other species), enabling systematic training and evaluation under diverse morphologies. Experimental results demonstrate that NEC achieves high-fidelity motion reconstruction with substantial compression, while effectively disentangling motion from skeletal structure. This capability supports a broad range of downstream tasks, including cross-species motion transfer, motion composition, denoising, generation (plug-and-play with *any* token-based generator; e.g., MoMask) and motion–text retrieval (via an OwO-based CLIP variant). By grounding motion representation in BVH animation while removing species-specific constraints, NEC establishes a principled framework for universal motion analysis and synthesis across varied morphologies.

## 1 Introduction

Generating dynamic 4D content is central to world models that must reason about, predict, and act within complex environments. While deformation-field approaches extend static 3D representations (meshes, NeRFs, 3D Gaussians) with temporal dynamics, they often suffer from geometric inconsistency, high computational cost, and limited user control Li et al. (2024). Moreover, many pipelines still reconstruct 4D from videos or video generative priors Cao et al. (2024); Ren et al. (2024), which hinders the learning of generalizable motion priors and structured abstractions.

Against this backdrop, skeleton-based modeling offers an interpretable, compact, and controllable interface for motion, and has driven impressive progress in human-centric 4D content Hong et al. (2022); Tevet et al. (2023). However, most frameworks are ill-suited for cross-embodiment applications: some rely on fixed human templates that do not generalize to diverse lifeforms Tevet et al. (2023); Zhang et al. (2024); others retarget to a *shared canonical skeleton* Wang et al. (2025), reducing expressivity for disparate body plans and bone-length statistics; and keypoint-only representations are misaligned with modern CG pipelines Guo et al. (2022a); Plappert et al. (2016). As a result, these designs implicitly assume near-static topology and bone lengths, limiting multi-species simulation, cross-embodiment animation, and universal motion understanding.

To move beyond fixed templates, we introduce a topology-agnostic motion representation that operates on *arbitrary* Biovision Hierarchy (BVH) skeletons. Concretely, we aggregate heterogeneous skeletal animations into a shared latent space via a unified tokenizer with three parts: first, an *ontology-aware skeletal graph encoder (OwO)* that embeds a rest-pose skeleton into per-joint descriptors by encoding joint-name semantics, rest-pose offsets, and graph topology; second,

a *Topology-Agnostic Tokenizer (TAT)* that discretizes BVH motion on arbitrary skeletons using structure-aware spatio-temporal blocks; and third, a new BVH-centric benchmark (*Unified BVH Universe, UvU*) spanning heterogeneous species and topologies. Together, these components enable cross-skeleton motion representation and broad downstream use.

The core challenge is to extract *topology-invariant yet semantics-preserving* motion codes that accommodate diverse joint hierarchies and spatio-temporal relations. To that end, we first train a graph encoder OwO in a self-supervised manner to map the rest pose into per-joint embeddings with auxiliary objectives that capture topological, postural, and semantic cues. We carefully design four loss functions to improve the representation capacity. Building on these embeddings, we then construct a spatio-temporal encoder–decoder that consumes OwO features and processes BVH sequences on arbitrary skeletons. In turn, the resulting tokenizer is plug-and-play for discrete token-based generators and generalizes natively across skeletons.

Turning to data, we consolidate three primary sources—HumanML3D Guo et al. (2022a), Objaverse-XL Deitke et al. (2023), and Truebones Zoo Truebones (n.d.)—through extensive cleaning, normalization, and annotation. By adopting BVH as the unified representation, the benchmark facilitates large-scale aggregation and remains compatible with production animation pipelines. In total, **Unified BVH Universe** contains **47,807** high-quality motion sequences with text, covering humans, quadrupeds, and other species, and enabling systematic evaluation under arbitrary topologies.

Finally, under a unified protocol, our tokenizer achieves strong reconstruction while offering substantial compression. It consistently outperforms the RVQVAE baseline in retrieval metrics, generation fidelity, and joint-space accuracy. Beyond reconstruction, the discrete token space supports *any-skeleton* motion transfer, composition, and token-level editing *by virtue of its token-level, skeleton-agnostic formulation*. Furthermore, the topology-agnostic token space is plug-and-play with *any* token-based generator (e.g., MoMask), enabling unified and flexible *any-skeleton* motion generation. Meanwhile, the OwO structural prior empowers *any-skeleton* Motion–Text CLIP alignment, providing a complementary interface for motion analysis and retrieval.

Our contributions are threefold:

1. **Ontology-aware Skeletal Graph Encoder.** We introduce a dedicated graph embedder and design four self-supervised loss functions, enabling pre-training on 3D rigging data to produce meta-representations for each keypoint. These representations encapsulate topological, postural, and semantic features, supporting diverse downstream tasks oriented toward BVH-format data.

2. **Topology-Agnostic Tokenizer.** We develop a graph-conditioned motion tokenizer that operates on arbitrary BVH skeletons and produces compact, token-level, skeleton-agnostic representations. This design enables a variety of additional appealing applications once the tokenizer has been trained.

3. **BVH-centric benchmark.** We establish a large-scale, curated BVH benchmark (47,807 sequences) spanning heterogeneous species and skeletal topologies. The benchmark supports standardized evaluation of topology generalization arbitrary for reconstruction, retrieval (R-Precision@K), and distributional quality (FID), while remaining compatible with production animation pipelines.

## 2 RELATED WORKS

**Skeleton-based Motion Generation.** Skeletons offer a compact, interpretable, and controllable interface for motion, and have progressed rapidly under text-, audio-, and scene-conditioned settings. Text-to-motion moved from embedding/variational alignment (MotionCLIP, TEMOS) to diffusion-based backbones with higher fidelity and local editing Tevet et al. (2022); Petrovich et al. (2022); Tevet et al. (2023); Kim et al. (2023). Large-scale controllability and efficiency were improved via multi-level conditioning and retrieval augmentation (MotionDiffuse, ReMoDiffuse), parameter-efficient adaptation (LoRA-MDM), and latent/AR decoding (MLD, DART) Zhang et al. (2024; 2023); Chen et al. (2023); Zhao et al. (2025). Audio-conditioned gesture and dance generation followed a similar trajectory, from deterministic or VAE-style mappings to diffusion/transformer

hybrids with rhythm- or key-pose guidance Li et al. (2021); Ao et al. (2023); Siyao et al. (2022); Huang et al. (2025). Despite strong results, most systems assume a fixed human topology (often with near-fixed bone lengths), which hinders transfer to non-human embodiments and limits cross-species generalization and retargeting under heterogeneous skeletons.

**Discrete Motion Tokenization.** A parallel line discretizes motion into compact token sequences to enable LLM-style generation, editing, and retrieval. Vector-quantized autoencoders Van Den Oord et al. (2017) (including residual/hierarchical variants) and masked modeling have been used to build motion codebooks Guo et al. (2023) and unify multiple tasks within one interface Jiang et al. (2024). Compared to continuous diffusion alone, discrete pipelines are amenable to long-horizon reasoning, scalable data mixing, and plug-and-play conditioning with language and audio. However, existing tokenizers are largely tied to a canonical human skeleton and a fixed joint set, limiting their use as a universal representation for diverse lifeforms Guo et al. (2022b). In contrast, we condition tokenization on an *ontology-aware skeletal graph* extracted from the rest pose, producing topology-agnostic BVH motion codes that generalize across arbitrary skeleton templates and remain compatible with discrete token-based generators.

**Cross-Embodiment, Retargeting, and BVH-centric Corpora.** Cross-species motion is commonly tackled by mapping motions to a canonical template (e.g., SMPL, GHUM) or by pairwise skeleton-aware retargeting networks Loper et al. (2015); Xu et al. (2020); Aberman et al. (2020). While effective for human avatars, such normalization sacrifices expressivity for disparate body plans and bone length statistics. Recent efforts extend beyond humans, exploring arbitrary topology or animal motion generation Gat et al. (2025); Wang et al. (2025), yet a universal tokenizer that natively supports arbitrary BVH templates is still underexplored. Given BVH's prevalence in mocap and digital content creation pipelines and its direct compatibility with animation tools and game engines, adopting BVH as a unifying substrate facilitates large-scale aggregation and standardized evaluation. We therefore release a BVH-centric corpus that consolidates heterogeneous sources (e.g., HumanML3D, Objaverse-XL and community mocap libraries) Guo et al. (2022a); Deitke et al. (2022; 2023); Truebones (n.d.), and we evaluate generalization under seen/unseen topologies and species to reflect real production settings.

## 3 UvU: UNIFIED BVH UNIVERSE

### 3.1 DATASET OVERVIEW

The Unified BVH Universe dataset is designed to enable generalized motion understanding and synthesis across varying skeletal topologies. Unlike existing datasets that focus on a single skeletal structure or similar structures (e.g., human-only Loper et al. (2015) or species-specific motions Biggs et al. (2020)), our dataset introduces highly divergent skeletons like fantasy creatures, enabling generative models with the ability to drive a variety of skeletons. The data format we are using within Unified BVH Universe, as shown in Fig. 1, (2) representation part, consists of 4 parts: **BVH Motion Data** for temporal joint rotations, **Base Mesh** for 3D mesh template, **Skin Weights** for mesh deformation and **Text Annotations**. In summary, our dataset consists of 47,807 animations.

### 3.2 DATA PREPROCESSING

As shown in Fig. 1, to construct the Unified BVH Universe dataset, we unify motion data from three heterogeneous sources: HumanML3D Guo et al. (2022a), Objaverse-XL Deitke et al. (2023), and Truebones Zoo Truebones (n.d.). For HumanML3D, we reconstruct animations from SMPLX Pavlakos et al. (2019) parameters, standardize skeleton structures, and convert them into BVH format. For Objaverse-XL, extensive filtering and correction are conducted to acquire high quality BVH animations. Through such preprocessing, we ensure that all animation data have only a single skeleton tree, global translations are aggregated on the root joint, and remove redundant or meaningless joints. Finally, Qwen2.5-VL (Bai et al., 2025) is utilized to further filter semantically acceptable motions and generate text annotations for them. Truebones Zoo provides diverse artist-animated FBX motions; we extract skeletal structures and animations, apply standardization, and supplement with human-annotated text descriptions. All data sources undergo a unified filtering and smoothing process to ensure temporal consistency and physical plausibility. After preprocessing, we split each

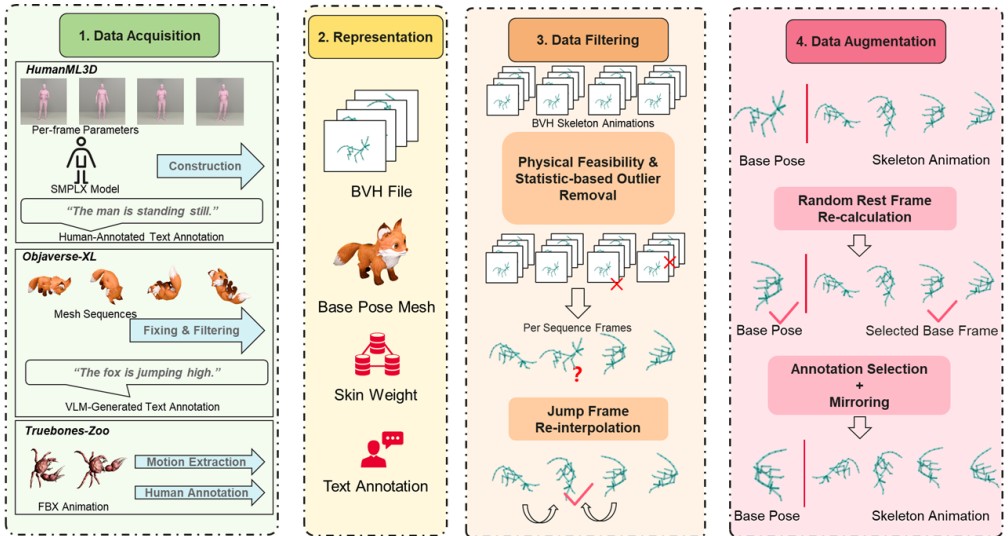

Figure 1: **Overview of Unified BVH Universe dataset pipeline.** We collect the original data from 3 existing datasets. Firstly, different motion data are unified to our standardized representations, consist of the BVH file, base pose mesh, skin weights and text annotations. Additionally, data filtering and smoothing are applied on the raw data to ensure the physical feasibility and correctness. Finally, during the training process, we utilize on-the-fly augmentations to further expand the diversity.

dataset separately. For HumanML3D, we use the original split Guo et al. (2022a). For Objaverse-XL, we randomly divide the sequences into 85% train and 15% test. For Truebones Zoo, we ensure that at least one animal per category (e.g. biped, quadruped) is included, with 15% of sequences as test and the rest as train. Detailed preprocessing procedures are provided in the supplementary material.

## 4 METHODS

To support arbitrary skeletal topologies, our tokenizer must incorporate graph-based reasoning. In our framework, we introduce a standalone Graph Embedder, which transforms the rest pose skeleton into node embeddings. These embeddings are then used in the encoder and decoder to distinguish the semantics of each joint. Since the graph representation only needs to be encoded once per skeleton, the graph structure can be discarded during subsequent tokenization and generation, significantly reducing computational cost.

An overview of the tokenizer structure is shown in Fig. 2. We next describe the architecture of the Graph Embedder and its integration into the encoder and decoder.

### 4.1 PROBLEM DEFINITION

We unify all animation data into the BVH motion format during the data processing stage. Specifically, an animation sequence with $J$ joints and $T$ frames can be represented as $\Theta \in \mathbb{R}^{T \times J \times 9}$. For the root joint ($j = 0$), the representation is defined as $\Theta_{t,0} = \begin{bmatrix} \Delta x_t & \Delta y_t & \Delta z_t & \text{rot6D}_t^{(0)} \end{bmatrix}$, where the first three dimensions encode the displacement relative to the previous frame along the three coordinate axes, and the last six dimensions denote the global orientation in rot6D (Zhou et al., 2019) format. For non-root joints ($j \neq 0$), the representation is $\Theta_{t,j} = \begin{bmatrix} 0 & 0 & 0 & \text{rot6D}_t^{(j)} \end{bmatrix}$, where the last six dimensions represent the relative rotation with respect to their parent node in the BVH kinematic tree, also expressed in rot6D format.

Note that all joint rotations are defined relative to the rest pose, as specified by the BVH skeleton hierarchy. As a result, encoding and decoding BVH motions inherently requires access to the underlying rest pose. Specifically, the rest pose includes a kinematic tree that encodes the parent–child

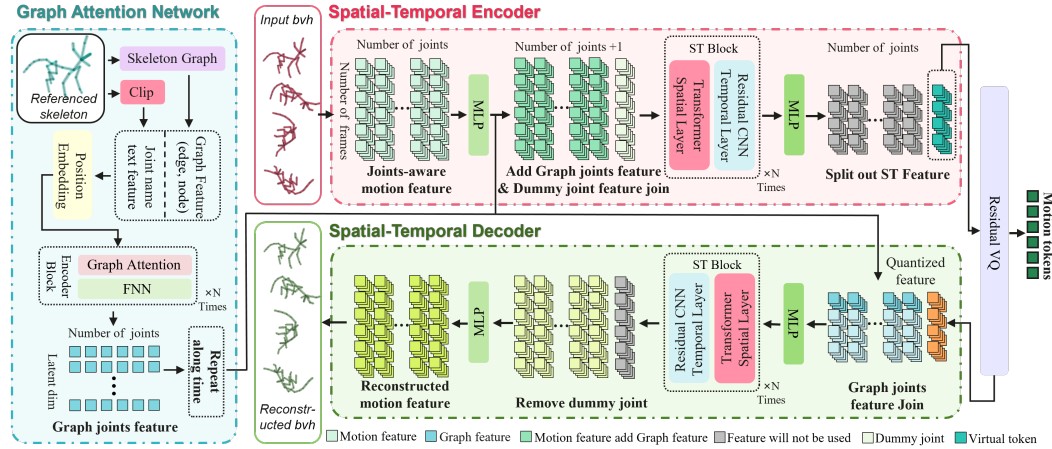

Figure 2: **Overview of NECromancer (NEC).** NEC consists of two main components: (a) Ontology-aware Skeletal Graph Encoder (OwO), which encodes static skeletal information (topology, joint names, rest pose) into structured graph-based joint features;(b) Topology-Agnostic Tokenizer (TAT), including Spatio-Temporal Encoder and Decoder, which maps motion sequences into a unified feature space, appends virtual joints, and converts them into discrete motion tokens.

relationships among joints, i.e., the topology $\mathcal{S} = (\mathcal{J}, \mathcal{E})$, where $\mathcal{J}$ is the set of joints and $\mathcal{E} \subseteq \mathcal{J} \times \mathcal{J}$ represents parent-child edges, as well as the offset of each joint $j$ with respect to its parent node $i$, denoted as $\mathbf{o}_{i \to j} \in \mathbb{R}^3$. Moreover, the rest pose also provides the name of each joint.

## 4.2 OwO: ONTOLOGY-AWARE SKELETAL GRAPH ENCODER

Since our tokenizer needs to handle different skeletal structures with varying numbers of joints, we aim to design a unified modeling scheme. Specifically, we require the tokenizer to transform the BVH motion $\Theta$ into a latent code

$$\mathbf{z} \in \{1, 2, \ldots, K\}^{\lfloor \frac{T}{r} \rfloor \times R},$$

where $r$ denotes the temporal compression ratio, $R$ is the number of residual tokens used in Residual Vector Quantization (RVQ) to represent the same continuous latent feature, and $K$ is the size of the codebook.

This formulation implies that, in the encoder stage of the tokenizer, spatial information across all joints within each frame must be effectively fused. In the decoder stage, the latent features must be able to accurately reconstruct the motion information of each joint. To achieve high-quality motion reconstruction, we require a robust embedding that can reliably distinguish each joint given its rest pose. Therefore, we design a encoder that leverages the full information of the rest pose to extract a unique feature for each joint, serving as its identity embedding. To effectively model the topological structure among joints in the BVH, we construct a graph based on the rest pose information, and build several graph attention blocks on top of it for feature extraction. The detailed computational formulas are provided in the appendix.

Since 4D data is extremely scarce, we design a pre-training stage to better train the graph encoder, which can be applied to arbitrary rigged 3D models. Specifically, we design a set of self-supervised objectives to guide the training of the Graph Embedder. These tasks are crafted to encourage the node and global features to encode three critical aspects of skeletal structure: **geometric**, **topological**, and **semantic** information. The module is pretrained independently and frozen during downstream motion generation to serve as a transferable structural prior.

For the extracted node features, we design the following three types of loss functions to encourage them to capture different aspects of information:

- **Geometric Loss.** This loss focuses on the recovery ability of the rest pose. For any pair of joints $(i, j)$ information and their corresponding node features, a task-specific prediction

head is required to output the offset of joint $j$ relative to joint $i$. If the model can accurately solve this task, then the relative positions of all joints in the rest pose can be recovered.

- **Topological Loss.** This loss targets the connectivity information encoded in the kinematic tree of the original BVH. Leveraging the tree structure, we require the model to correctly identify the lowest common ancestor (LCA) of any joint pair $(i, j)$ after the task-specific prediction head.

  **Theorem.** *If a model can correctly determine the LCA for any pair of nodes $(i, j)$ in a tree, then the entire tree topology can be uniquely reconstructed.*

- **Semantic Loss.** This loss encourages alignment between node features and semantic information. We extract textual features of each joint name using CLIP, and apply a contrastive learning objective such that each joint's node feature is pulled closer to its own name embedding, while being pushed away from those of other joints.

The detailed loss calculation and the proof of theorem are thoroughly introduced in the appendix.

**Role of OwO within the Tokenizer.** OwO is not an auxiliary component but the structural prior that enables topology-agnostic motion tokenization. Given a skeleton, OwO produces a set of joint-level identity embeddings

$$F_{\text{node}} = \{h_j \in \mathbb{R}^d\}_{j=1}^J,$$

which encode the semantic, geometric, and topological roles of all joints.

During TAT encoding, these structural embeddings are fused with the per-joint motion features:

$$X_{t,j} = \text{MLP}(\Theta_{t,j}) + \text{Proj}(h_j),$$

ensuring that the tokenizer interprets motions with respect to the correct anatomical meanings. This fusion allows the model to consistently understand motion patterns such as *arm lifting*, *spine bending*, or *wing flapping*, even when applied to skeletons with different numbers of joints or distinct hierarchical structures.

Importantly, OwO also plays a crucial role during decoding. Given a target skeleton, its OwO embeddings $F_{\text{node}}$ are repeated across the temporal dimension and concatenated to the quantized latent sequence:

$$\tilde{Z}_{t,j} = [\, z_t \,\|\, h_j \,],$$

where $z_t$ is the virtual-joint latent token at timestep $t$. This provides the decoder with a skeleton-specific *template*, enabling it to reconstruct per-joint rotations consistent with the target morphology while preserving the motion dynamics encoded in the token sequence. Since OwO is computed once per skeleton and reused throughout the entire TAT pipeline, it serves as a lightweight yet expressive structural descriptor for universal motion reconstruction.

### 4.3 TAT: TOPOLOGY-AGNOSTIC TOKENIZER

**Motivation: Why Topology-Agnostic Tokenization Is Challenging.** Conventional VQ-based motion tokenizers quantize the feature at every joint and timestep, which implicitly assumes a fixed skeleton layout. This requirement fundamentally prevents them from handling heterogeneous skeletons with different joint counts and joint orderings, such as humans (22 joints), dogs (87 joints), birds (with folding wings), or dragons (over 120 joints).

To support motion tokenization across arbitrary skeletons, the quantization stage must be decoupled from the number of joints. Therefore, instead of quantizing joint-wise features, we introduce a *virtual joint* that summarizes all joint features at each timestep into a topology-invariant representation. This design removes the dependence on the underlying kinematic structure and enables a truly universal discrete motion space.

**Virtual Joint for Topology-Invariant Quantization.** Instead of aggregating joint features via pooling, we introduce a learnable *virtual joint* token, analogous to a classification token in Transformer encoders. At each (downsampled) timestep $t$, we augment the set of joint features $\{X_{t,j}\}_{j=1}^J$ with a virtual joint embedding $v_t^{(0)} \in \mathbb{R}^d$:

$$\tilde{X}_t = \{X_{t,1}, \ldots, X_{t,J}, v_t^{(0)}\}.$$

This extended sequence is processed by $L$ stacked spatio-temporal blocks. Through the spatial attention, the virtual joint attends to all real joints and gradually accumulates a global summary of the motion at timestep $t$. We denote the virtual joint after the last block as $v_t^{(L)}$.

Crucially, only the virtual joint is fed into the RVQ quantizer:

$$z_t = \text{RVQ}(v_t^{(L)}),$$

which yields a fixed-size discrete latent code independently of the number of real joints. During decoding, the quantized latent $z_t$ is injected back as the virtual joint token and combined with the OwO-conditioned joint features of the target skeleton to reconstruct per-joint rotations. This CLS-style virtual joint design enables topology-invariant motion tokenization without ever requiring a fixed joint grid.

Thanks to the explicitly extracted skeletal structure information from the Graph Embedder, we can simplify the motion reconstruction process under arbitrary topologies. Specifically, the inputs to this reconstruction step include:

- The node-level structural embeddings from the Graph Embedder, denoted as $\mathbf{F}_{\text{node}} = \{\mathbf{h}_j \in \mathbb{R}^d\}_{j=1}^J$, where $J$ is the number of joints and $d$ is the embedding dimension;

- A motion sequence in our defined format, represented as $\boldsymbol{\Theta} \in \mathbb{R}^{T \times J \times 9}$, where $T$ is the number of frames and each 9D vector encodes joint translation/rotation information as defined in Section 4.1.

By combining $\mathbf{F}_{\text{node}}$ with the per-frame motion features in $\Theta$, we can reconstruct motion tokens in a topology-agnostic manner without requiring explicit graph traversal during generation.

**Difference from Prior Spatio-Temporal Transformers.** Unlike prior motion Transformers that operate on fixed human skeletons, the proposed TAT introduces three key innovations:

- **Graph-conditioned spatial attention.** Each joint feature is modulated by its OwO embedding, allowing the spatial attention module to reason over anatomical semantics rather than relying solely on joint indices.

- **Topology-invariant quantization via the virtual joint.** Only the virtual joint is quantized, enabling a discrete representation that is independent of the number or ordering of joints.

- **Decoupled structure–motion representation.** The motion tokens contain no structural information; the structure is injected only through OwO at encode and decode time. This makes the latent motion code universally applicable to any skeleton.

Together, these innovations make TAT the first spatio-temporal tokenizer capable of reconstruction, transfer, and generation across arbitrary skeletal topologies.

**Spatio-temporal Modeling.** As shown in Figure 2, the encoder contains $L$ spatio-temporal blocks. Each block consists of: A **temporal module**, comprising a 1D convolution (with stride $s$) and a ResNet1D block for temporal abstraction; A **spatial transformer**, which models joint interactions at each timestep using multi-head self-attention.

At each layer, temporal downsampling reduces the sequence length by a factor of $s$, resulting in an overall downsampling rate $r = s^L$. The final feature is reshaped and passed through a $1 \times 1$ convolution to produce quantized features $\mathbf{Z}_{\text{feat}} \in \mathbb{R}^{T/r \times J \times W}$, from which token quantization is performed. The virtual joint token is extracted separately and used as a global summary feature.

The decoder mirrors the encoder structure in reverse. Given a quantized token sequence and the corresponding joint features, it performs: Concatenation of per-joint features and the global token; Spatial transformer operations for joint-wise refinement; Upsampling and reverse-dilated temporal convolutions to restore full temporal resolution.

The virtual joint token is removed before output. The decoder maps the output back to the original motion space $\mathbb{R}^{T \times J \times 9}$ using a linear projection. This design allows the tokenizer to encode complex spatio-temporal patterns in a structure-aware yet data-efficient manner, and produce discrete tokens suitable for downstream generative modeling. Implementation details can be found in the supplementary materials.

Table 1: **Reconstruction Results on three different datasets.**

| Method | MPJPE ↓ | | | MPJPE (no trans) ↓ | | | GeoDist ↓ | | |
|---|---|---|---|---|---|---|---|---|---|
| | H3D | Obj-XL | Zoo | H3D | Obj-XL | Zoo | H3D | Obj-XL | Zoo |
| T2M-GPT | 0.4203 | 0.2583 | 0.2271 | 0.1376 | 0.2128 | 0.0843 | 6.84° | 28.66° | 18.76° |
| Motion Streamer | 0.1961 | 0.2456 | 0.2261 | 0.0972 | 0.1963 | 0.0842 | 5.37° | 26.17° | 18.73° |
| TM2T | 0.1411 | 0.1918 | 0.1434 | 0.0873 | 0.1565 | 0.0757 | 5.34° | 21.49° | 17.28° |
| RVQVAE (zero pad) | 0.4729 | 0.2228 | 0.1762 | 0.2688 | 0.1817 | 0.1143 | 27.95° | 22.72° | 18.76° |
| **NEC w/ VQ** | 0.3960 | 0.1840 | 0.1657 | 0.1395 | 0.1343 | 0.0828 | 7.78° | 17.40° | 16.19° |
| **NEC w/ RVQ** | **0.1084** | **0.0983** | **0.1008** | **0.0588** | **0.0787** | **0.0635** | **3.96°** | **12.12°** | **13.88°** |

## 4.4 DATA AUGMENTATION

To enhance topological generalization, we propose a base pose randomization technique that preserves semantic content while expanding motion diversity. The method involves: (1) selecting a random frame as new rest pose, (2) computing global rotations and positions, (3) deriving new rest pose offsets, and (4) recalculating local rotations using key equations (see Appendix C.1).

Our data augmentation generates physically plausible animations with semantically consistent motions, encouraging the model to prioritize semantic meaning and physical correctness.

## 5 EXPERIMENTS

### 5.1 EXPERIMENTAL SETUP

**Datasets and splits.** We evaluate on three BVH-centric sources: **HumanML3D** Guo et al. (2022a), **Objaverse-XL** Deitke et al. (2023), and **Truebones Zoo** Truebones (n.d.). Unless otherwise noted, we train *one* model on the *union* of training partitions from all three sources (*UvU-train*) and evaluate on the *union* of their test partitions (*UvU-test*); per-dataset numbers are then computed by restricting evaluation to the corresponding source's test subset. All methods (baselines and ours) are trained and evaluated under this unified protocol.

**Baselines.** We compare against canonical **VQVAE** Van Den Oord et al. (2017) and **RVQVAE** Guo et al. (2023) tokenizers trained on padded/masked sequences to a max-joint layout (single canonical skeleton). Our method **NEC** combines the **OwO** (ontology-aware skeletal graph encoder) with the **TAT** tokenizer to produce topology-agnostic motion tokens on arbitrary BVH skeletons.

**Implementation details.** Unless stated, OwO uses 8 graph attention blocks with 512 latent dimension while TAT uses 3 spatio-temporal blocks. Each spatio-temporal block contains 2 convolution layers and 2 spatial transformer encoder layers. The used quantizer is a 6-layer codebooks (size 1024). We train with AdamW, cosine LR, gradient clipping, and random rest-pose augmentation (Sec. 4). The tokenizer is trained with 32 Ascend 910. The whole training process contains around 24k iterations. Initial learning rate is 2e-4 and is decreased to 2e-5 during the last 4k iterations.

**Metrics.** We report four metrics throughout: **MPJPE** ↓ (root-aligned, joint-set-agnostic), **FID** ↓ (computed on the same retrieval backbone as prior work), **GeoDist** ↓ (mean geodesic distance of joint rotations), and **R-Precision@$\{1, 2, 3\}$** ↑ for text–motion retrieval.

**Other applications of Graph Embedder.** For understanding BVH motion, accurately extracting features for each joint is crucial. Beyond the tokenizer, we also experimented with two contrastive learning models—PoseVAE and Text–Motion Evaluator—to validate the effectiveness of our proposed Graph Embedder. Both models share a similar overall structure with the tokenizer, but with key differences: PoseVAE operates on single frames and employs a VAE for modeling, whereas the Evaluator applies a transformer along the temporal dimension to extract a unified feature from the motion sequence, which is then used to compute similarity with textual representations.

Table 2: Ablation study on pretraining and different loss settings. We report results on three tasks and three datasets. Lower is better for MPJPE, higher is better for R Precision Top 1.

| Pretrain | Loss Setting | | | | Pose VAE (MPJPE) | | | Motion VQVAE (MPJPE) | | | Evaluator (R Precision) | | |
|---|---|---|---|---|---|---|---|---|---|---|---|---|---|
| | Offset | LCA | Dist. | Con. | H3D | Obj-XL | Zoo | H3D | Obj-XL | Zoo | H3D | Obj-XL | Zoo |
| No | 0 | 0 | 0 | 0 | 0.0940 | 0.0618 | 0.0599 | 0.1458 | 0.1034 | **0.0995** | 0.5355 | 0.1655 | 0.2756 |
| Yes | 1 | 0 | 0 | 0 | 0.0443 | 0.0442 | 0.0485 | 0.1151 | 0.1028 | 0.1122 | 0.4052 | 0.1310 | 0.2677 |
| Yes | 0 | 1 | 0 | 0 | 0.0530 | 0.0514 | 0.0585 | 0.1300 | 0.1051 | 0.1529 | 0.3913 | 0.1448 | 0.2913 |
| Yes | 0 | 0 | 1 | 0 | 0.0623 | 0.0412 | 0.0466 | 0.1138 | 0.1024 | 0.1146 | 0.3731 | 0.1793 | **0.3150** |
| Yes | 0 | 0 | 0 | 1 | 0.0399 | 0.0502 | **0.0433** | 0.1126 | 0.1050 | 0.1006 | 0.5215 | 0.1172 | 0.2913 |
| Yes | 0 | 1 | 1 | 1 | 0.0822 | **0.0409** | 0.0485 | 0.1084 | **0.0983** | 0.1008 | **0.5713** | **0.2207** | 0.2992 |
| Yes | 1 | 0 | 1 | 1 | 0.0627 | 0.0424 | 0.0475 | **0.1073** | 0.1017 | 0.1359 | 0.5604 | 0.1586 | 0.3071 |
| Yes | 1 | 1 | 0 | 1 | 0.0780 | 0.0410 | 0.0460 | 0.1343 | 0.1098 | 0.1058 | 0.5580 | 0.1586 | 0.2283 |
| Yes | 1 | 1 | 1 | 0 | **0.0363** | 0.0411 | 0.0521 | 0.1567 | 0.1074 | 0.1003 | 0.5434 | 0.1793 | 0.2835 |
| Yes | 1 | 1 | 1 | 1 | 0.1009 | 0.0516 | 0.0661 | 0.1147 | 0.1088 | 0.1144 | 0.4976 | 0.1793 | 0.2992 |

## 5.2 MAIN RESULTS: RECONSTRUCTION

**Summary.** Table 1 summarizes reconstruction performance across three heterogeneous datasets: HumanML3D (human motions), Objaverse-XL (large-scale multi-object motions with diverse rigs), and Truebones Zoo (artist-animated non-human skeletons). For T2M-GPT, Motion Streamer, TM2T, and the traditional RVQVAE baseline, we pad all skeletons to the fixed joint layout required by these models, ensuring compatibility with their human-centric architectures.

Across all three datasets, a consistent trend emerges. Human-centric models (T2M-GPT, Motion Streamer, TM2T) perform reasonably on HumanML3D but degrade significantly on Objaverse-XL and Zoo, where joint counts, limb structures, and bone-length statistics differ widely. This shows that models assuming fixed human topology cannot generalize to heterogeneous skeletons. Padding-based RVQ-VAE baselines perform even worse, with high MPJPE(no-trans) and GeoDist indicating that zero-padding introduces structural ambiguity and prevents accurate rotational reconstruction.

By contrast, NEC overcomes these limitations through an ontology-aware structural prior (OwO) and topology-invariant quantization (TAT). Even NEC w/ VQ surpasses all padding-based VQ baselines, showing the benefits of structure-aware encoding. The full NEC w/ RVQ achieves the strongest results across all datasets and metrics, reducing MPJPE on HumanML3D by nearly 2× compared to TM2T, and delivering large improvements on Obj-XL and Zoo. Its consistently lowest GeoDist further confirms superior rotational accuracy and cross-topology fidelity.

Overall, the results demonstrate that topology-aware conditioning combined with residual quantization is crucial for precise and structurally consistent motion reconstruction across diverse skeletal morphologies.

## 5.3 ABLATIONS

**OwO pretraining: reconstruction + retrieval (Table 2).** We ablate the effect of **OwO** pretraining by toggling its auxiliary objectives: Adjacent Offset regression (Offset), LCA/topology prediction (LCA), Distance regression on any pairs (Dist.), and name-text contrastive learning (Con.). Here Offset is a special case of Dist.

**Findings.** (1) Overall, combining the three loss functions—*LCA*, *Dist.*, and *Con.*—achieves the best performance across all tasks. (2) The *Offset* task, which predicts offsets between neighboring joints, is relatively simple and provides limited benefit for representation learning; in fact, it may even have a negative effect. (3) The *LCA* task substantially improves performance on the Zoo dataset, likely due to its highly diverse skeletal structures, indicating that learning topology enhances cross-skeleton generalization.

## 5.4 QUALITATIVE RESULTS

**Reconstruction vs. baselines.** Fig. 3 shows side-by-side BVH pose sequences for NEC, VQVAE, and RVQVAE on HumanML3D/Objaverse-XL/Truebones. NEC better preserves limb phasing and contact timing, especially on non-human skeletons.

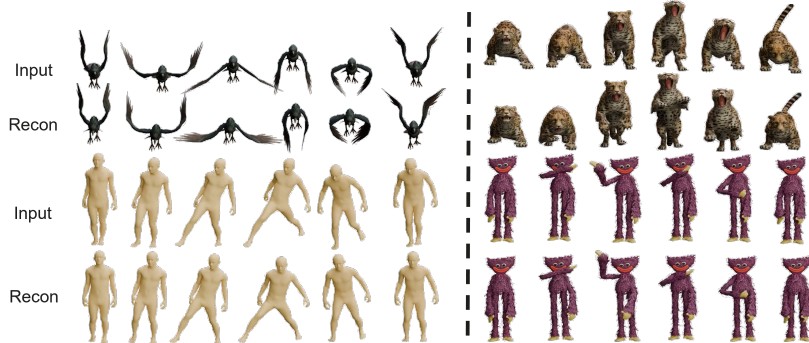

Figure 3: **Qualitative reconstruction.** NEC vs. GT on Objaverse-XL, and Truebones.

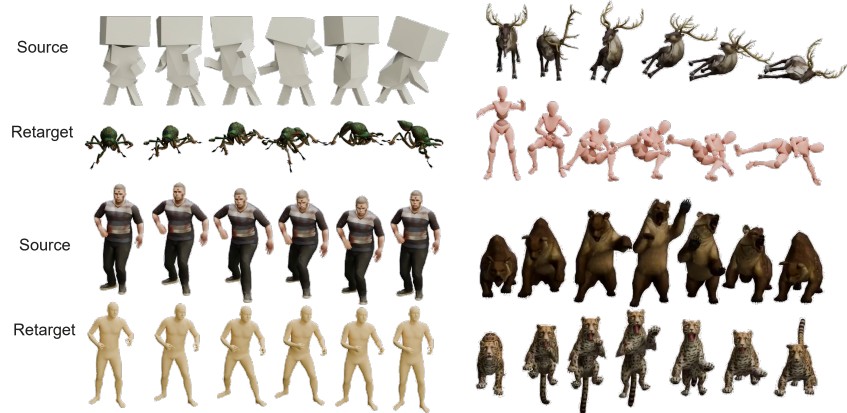

Figure 4: **Visualization of Motion Transfer examples.**

**Motion transfer through different species.** We demonstrate **motion transfer** (source tokens + target OwO) across arbitrary topologies (Fig. 4). *Crucially, this requires no task-specific fine-tuning or auxiliary heads*: because NEC produces *topology-agnostic* tokens and the decoder is *OwO-conditioned*, a single trained model supports **zero-shot transfer** by decoding source token sequences under different target OwOs (morphology swap). This enables direct motion transfer between species with diverse skeletons while maintaining temporal coherence.

**Plug-and-play generation.** We plug **NEC** tokens into a token-based generator (e.g., MoMask) for **any-skeleton** text-to-motion), using only target OwO to condition morphology. *More generally, any token-based generator*—autoregressive, masked modeling, *can be trained or fine-tuned directly on NEC tokens* without architectural changes: the generator treats NEC codes as its discrete vocabulary, while morphology is provided by OwO at decode time, yielding *any-skeleton* generation capacity. Please refer to the demo video for the generation results.

## 6 CONCLUSIONS

We presented a unified framework for motion representation learning with three key contributions. First, the *Ontology-aware Skeletal Graph Encoder* leverages graph embeddings and self-supervised losses to pre-train on rigged 3D data, yielding meta-representations that capture topology, posture, and semantics. Second, the *Topology-Agnostic Tokenizer* converts arbitrary BVH skeletons into compact, skeleton-agnostic tokens, enabling diverse downstream applications. Third, we established a large-scale *BVH-centric benchmark* (47,807 sequences) covering heterogeneous species and skeletons, supporting standardized evaluation for reconstruction, retrieval, and distributional quality. These contributions together provide a foundation for topology-generalized motion modeling and evaluation in BVH format.

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

# APPENDIX FOR NECROMANCER

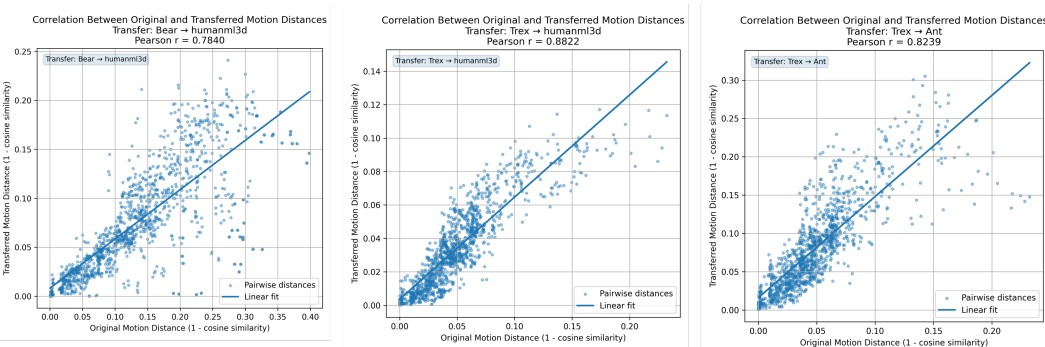

Figure 5: Motion Transfer Corelation Results for Rebuttal.

## REPRODUCIBILITY STATEMENT

We have made extensive efforts to ensure the reproducibility of NECromancer (NEC). The construction of the Unified BVH Universe (UvU) dataset is described in Sec. 3. The Ontology-aWare Skeletal Graph EncOder (OwO) and the Topology-Agnostic Tokenizer (TAT), as well as our data augmentation strategy, are presented in Sec. 4.2 and Sec. 4.3. Implementation details, including training procedures and hyperparameter settings, are provided in Sec. 5.1, while comprehensive descriptions of the model architectures are included in Appendix. C. To further support reproducibility, the data preprocessing code used for building UvU will be released in an anonymous GitHub repository upon paper acceptance.

## LLM USAGE

A large language model was used as a writing assistant to improve the clarity and readability of the manuscript. In addition, a vision-language model was employed to assist in data preprocessing and filtering. All final research decisions, data selection criteria, and methodological contributions were made and verified by the authors.

## A    DETAILS OF BVH REPRESENTATION

To enable motion generation across diverse skeletal structures, we adopt BVH (Biovision Hierarchy) as the unified representation format for all skeleton-based animations in our framework. BVH is a widely used hierarchical motion representation format that encodes both the skeletal topology (rest pose) and temporal motion trajectories in a compact and interpretable structure, as shown in Fig. 6. Its compatibility with commercial animation software and motion datasets makes it an ideal choice for bridging learning-based motion generation with practical deployment.

In this section, we provide a detailed explanation of the two critical components of the BVH format: the rest pose, which defines the static skeleton configuration, and the motion representation, which captures dynamic joint transformations over time.

### A.1    REST POSE

The rest pose in a BVH file encodes the skeleton hierarchy, which defines the parent-child relationship between joints, as well as the offset vectors between them. Each joint is represented by its name, position relative to its parent, and its degrees of freedom (DOF), such as rotation along the X, Y, and Z axes. The hierarchy is generally rooted at the Hips joint or an equivalent base (e.g., the Pelvis joint), and traverses downward through limbs and extremities.

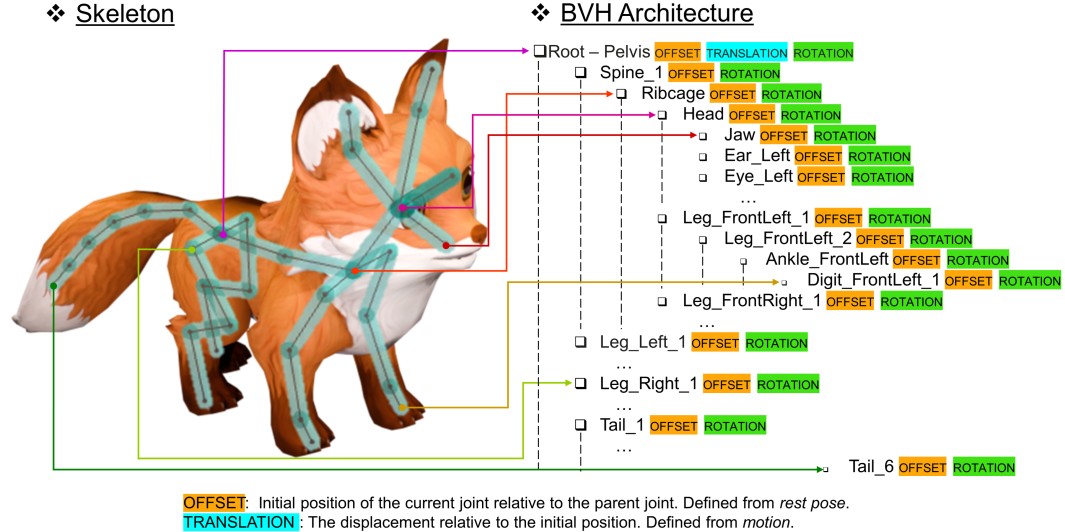

Figure 6: **Overview of BVH data structure.** The BVH motion format defines skeletal motion through a hierarchical structure and time-series data. The ROOT node, representing the base of the skeleton, includes an OFFSET field specifying its initial XYZ position in world coordinates . Subsequent JOINT nodes define offsets relative to their parent segments, establishing limb lengths and orientations. Each node also lists CHANNELS (the TRANSLATION and ROTATION in Figure) for root positional (only root node has TRANSLATION) and rotational (Z-X-Y order) data. The motion section includes frame counts, sampling rates, and per-frame data containing root translations(TRANSLATION) and joint rotations(ROTATION) applied in hierarchical order.

Importantly, the rest pose serves as the topological and spatial reference for interpreting all subsequent motion data. In our framework, we treat the rest pose as a graph, where nodes correspond to joints and edges reflect the skeletal hierarchy. This representation allows us to extract structural embeddings for each skeleton, facilitating the generalization of motion generation across arbitrary topologies. To ensure consistency across heterogeneous sources, we normalize all skeletons to a standard coordinate system and unify joint naming with convention.

### A.2 Motion Representation

The motion section of a BVH file records the transformations per frame applied to each joint over time. Each frame typically contains a set of scalar values corresponding to the DOF specified in the rest pose, usually in the form of Euler angles and root joint translations. These values are organized in a flat sequence for each time step, but semantically correspond to articulated motion governed by the skeletal hierarchy.

To handle skeletons of varying topology and DOF, we dynamically build the hierarchy graph for each BVH file based on its rest pose. This ensures that all motion sequences, regardless of their original structure, can be consistently represented and processed under a unified format.

## B Details of Data Preprocessing

To ensure a consistent and physically meaningful skeletal representation across the heterogeneous HumanML3D, Truebones Zoo and Diffusion4D datasets, we apply a unified and carefully designed preprocessing pipeline.

**(1) Correction of invalid or awkward root joint definitions.** Several skeletons in both datasets adopt unconventional root placements (e.g., a dummy root joint set outside of the subject body, or root incorrectly set to a shoulder or spine joint). Such configurations violate common skeletal conventions and introduce instability for models relying on hierarchical transformations. We identify

these problematic cases and relocate the root to an anatomically meaningful position (typically the pelvis), reconstructing the hierarchy so that all parent–child relationships follow consistent human and animal body kinematics.

**(2) Reassignment of global translations to the root joint.** In the raw data, some sequences embed joint-level translations on intermediate nodes, leading to frame-dependent bone-length drift and violating rigid-body kinematic constraints. To restore physical consistency, we transfer all global motion to the root joint and remove per-joint translations from all other nodes. This guarantees that bone lengths remain constant across frames and that only the root carries global displacement, while the rest of the skeleton expresses purely rotational motion.

**(3) Removal of sequences with abnormal bone lengths.** A subset of sequences contains extremely elongated or corrupted bones due to tracking errors or incorrect parameter export. Since such cases break kinematic plausibility and cannot be reliably corrected without strong assumptions, we exclude sequences whose bone lengths exceed a dataset-dependent threshold relative to their normalized diameter, effectively filtering out physically invalid examples.

**(4) Semantic standardization of joint names using a vision-language model.** Joint names provided in these datasets are highly inconsistent and often ambiguous (e.g., "joint1", "bone_02", "arm.L", or dataset-specific naming conventions). To achieve semantic uniformity, we employ a vision-language model (VLM) to map each joint name to a standardized human-skeleton vocabulary. This step not only harmonizes naming across datasets, but also ensures consistent semantic interpretation for downstream tasks such as retargeting, pose comparison, and learned conditioning.

**(5) Scale normalization based on skeletal diameter.** The raw datasets contain skeletons defined at widely different scales, resulting in inconsistent geometry and bone-length statistics. For each sequence, we compute the skeleton diameter, defined as the length of the longest kinematic chain from the root to any leaf joint. This measure offers a robust scale descriptor that is less sensitive to local bone-length noise. We uniformly scale the entire sequence using this diameter, ensuring that all skeletons across datasets share comparable global scale while preserving their relative proportions.

After applying the above stages, every sequence is represented by a single, coherent skeletal tree with fixed bone lengths, a unified joint naming scheme, and global motion encoded exclusively at the root. All redundant, duplicated, or semantically meaningless joints are removed. This preprocessing ensures topological and geometric consistency across datasets and enables fair and stable learning for all subsequent modules.

## C IMPLEMENTATION DETAILS

### C.1 DATA AUGMENTATION: MATHEMATICAL FORMULATIONS

The proposed data augmentation method follows these mathematical formulations:

1. Global rotation calculation:
$$\mathbf{r}_{t_0,i}^{\text{global}} = \mathbf{r}_{t_0,\text{parent}[i]}^{\text{global}} \cdot \mathbf{r}_{t_0,i}^{\text{local}} \tag{1}$$

2. Global position calculation:
$$\mathbf{p}_{t_0,i}^{\text{global}} = \mathbf{p}_{t_0,\text{parent}[i]}^{\text{global}} + \mathbf{r}_{t_0,\text{parent}[i]}^{\text{global}} \cdot \mathbf{o}_i \tag{2}$$

3. Rest pose offset derivation:
$$\bar{\mathbf{o}}_i = \mathbf{p}_{t,i}^{\text{global}} - \mathbf{p}_{t,\text{parent}[i]}^{\text{global}} \tag{3}$$

4. Local rotation re-calculation:
$$\bar{\mathbf{r}}_{t,i}^{\text{local}} = \left(\bar{\mathbf{r}}_{t,\text{parent}[i]}^{\text{global}}\right)^{-1} \cdot \mathbf{r}_{t,i}^{\text{global}} \cdot \left(\mathbf{r}_{t_0,i}^{\text{global}}\right)^{-1} \tag{4}$$

With such data augmentation process, we synthesize a large amount of animations with reasonable base poses, which is more physically plausible compared to random rotation jittering. By training on many animations that are different in numeric representations but the same on semantic meanings, the model is strongly encouraged to focus more on the semantic meaning and physical correctness of the animations.

## C.2   GRAPH EMBEDDER

### C.2.1   GRAPH INITIALIZATION

Given a rest pose skeleton $\mathcal{S} = (\mathcal{J}, \mathcal{E})$, where $\mathcal{J}$ is the set of joints and $\mathcal{E} \subseteq \mathcal{J} \times \mathcal{J}$ represents parent-child edges, we construct a directed graph where each joint is treated as a node.

To enable graph-based message passing, we initialize three types of features: node features, edge features, and a special global node.

### C.2.2   NODE FEATURE INITIALIZATION

Each joint $j \in \mathcal{J}$ is annotated with a semantic name. We use a pretrained CLIP text encoder to extract a 512-dimensional embedding $\mathbf{e}_j^{\text{text}} \in \mathbb{R}^{512}$, which is projected to the graph dimension:

$$\mathbf{h}_j^{(0)} = \text{FC}\left(\mathbf{e}_j^{\text{text}}\right) \in \mathbb{R}^d. \tag{1}$$

### C.2.3   GLOBAL NODE INITIALIZATION

We introduce a special node with name "global" and extract its CLIP embedding in the same manner:

$$\mathbf{h}_{\text{global}}^{(0)} = \text{FC}\left(\mathbf{e}_{\text{global}}^{\text{text}}\right) \in \mathbb{R}^d. \tag{2}$$

The global node is connected bidirectionally to all other joints:

$$\mathcal{E}_{\text{global}} = \left\{ (\text{global} \to j), \ (j \to \text{global}) \mid j \in \mathcal{J} \right\}.$$

These edges are treated identically to regular edges during attention, allowing the global node to gather holistic context from the graph while also distributing high-level signals.

### C.2.4   EDGE FEATURE INITIALIZATION

For each directed edge $(i \to j) \in \mathcal{E} \cup \mathcal{E}_{\text{global}}$, we define a bidirectional edge feature.

**Forward edge:**

$$\mathbf{e}_{i \to j}^{(0)} = \text{FC}\left( \left[ \mathbf{h}_i^{(0)} \parallel \mathbf{h}_j^{(0)} \parallel \phi(\mathbf{o}_{i \to j}) \right] \right), \tag{3}$$

**Backward edge:**

$$\mathbf{e}_{j \to i}^{(0)} = \text{FC}\left( \left[ \mathbf{h}_j^{(0)} \parallel \mathbf{h}_i^{(0)} \parallel \phi(-\mathbf{o}_{i \to j}) \right] \right), \tag{4}$$

where $\mathbf{o}_{i \to j} \in \mathbb{R}^3$ is the offset from joint $i$ to joint $j$, $\parallel$ denotes the concatenation operation, and $\phi(\cdot) \in \mathbb{R}^{3d}$ is a sinusoidal embedding per dimension.

For global-to-joint edges, we set $\mathbf{o}_{i \to j} = \mathbf{0}$, but retain the learned joint names for each endpoint.

### C.2.5   GRAPH ENCODER BLOCK

The graph is processed by $L$ stacked Graph Encoder Blocks, each consisting of a Graph Attention Layer and a Feed-Forward Network (FFN).

### C.2.6   GRAPH ATTENTION LAYER

Let $\mathbf{h}_j^{(l)}$ be the feature of node $j$ at layer $l$. We compute:

$$\tilde{\mathbf{h}}_j^{(l+1)} = \sum_{k \in \mathcal{N}(j)} \alpha_{k \to j}^{(l)} \cdot \mathbf{v}_{k \to j}^{(l)}, \tag{5}$$

where $\alpha_{k \to j}^{(l)} \in [0, 1]$ is the attention weight, and $\mathbf{v}_{k \to j}^{(l)} \in \mathbb{R}^d$ is the edge-conditioned message:

$$\alpha_{k \to j}^{(l)} = \frac{\exp\left( \text{LeakyReLU}\left( \mathbf{a}^\top \left[ \mathbf{W}_{\text{node}} \mathbf{h}_j^{(l)} \parallel \mathbf{W}_{\text{node}} \mathbf{h}_k^{(l)} \parallel \mathbf{e}_{k \to j}^{(l)} \right] \right) \right)}{\sum_{k' \in \mathcal{N}(j)} \exp\left( \text{LeakyReLU}\left( \mathbf{a}^\top \left[ \mathbf{W}_{\text{node}} \mathbf{h}_j^{(l)} \parallel \mathbf{W}_{\text{node}} \mathbf{h}_{k'}^{(l)} \parallel \mathbf{e}_{k' \to j}^{(l)} \right] \right) \right)}, \tag{6}$$

$$\mathbf{v}_{k\to j}^{(l)} = \mathbf{W}_v \left[ \mathbf{h}_k^{(l)} \| \mathbf{e}_{k\to j}^{(l)} \right]. \tag{7}$$

$$\tilde{\mathbf{e}}_{k\to j}^{(l+1)} = \mathbf{e}_{k\to j}^{(l)} + \mathbf{W}_{\text{src}} \mathbf{W}_{\text{node}} \mathbf{h}_j^{(l)} + \mathbf{W}_{\text{tgt}} \mathbf{W}_{\text{node}} \mathbf{h}_k^{(l)}. \tag{8}$$

### C.2.7 FEED-FORWARD NETWORK

We apply an FFN to each node independently:

$$\mathbf{z}_j^{(l+1)} = \text{ReLU}\left( \mathbf{W}_2 \cdot \text{ReLU}(\mathbf{W}_1 \cdot \tilde{\mathbf{h}}_j^{(l+1)} + \mathbf{b}_1) + \mathbf{b}_2 \right), \tag{9}$$

$$\mathbf{h}_j^{(l+1)} = \text{LayerNorm}\left( \mathbf{h}_j^{(l)} + \mathbf{z}_j^{(l+1)} \right), \tag{10}$$

where $\mathbf{W}_1 \in \mathbb{R}^{d \times d_{\text{ff}}}, \mathbf{W}_2 \in \mathbb{R}^{d_{\text{ff}} \times d}$.

We perform similar transformations to obtain the updated edge feature $\mathbf{e}_{k\to j}^{(l+1)}$ from $\tilde{\mathbf{e}}_{k\to j}^{(l+1)}$.

### C.3 TRAINING OBJECTIVES OF GRAPH EMBEDDER

The detail of our proposed three categories of self-supervision tasks are listed as below:

- **Geometric Task — Distance Regression.** For each pair of joints $(j, k)$, we regress the offset vector from joint $j$ to joint $k$. Let $\mathbf{h}_j \in \mathbb{R}^d$ and $\mathbf{h}_k \in \mathbb{R}^d$ be the node embeddings after the final graph encoder layer. We predict:

$$\hat{\mathbf{o}}_{jk} = \mathbf{W}_{\text{geo}} \cdot \text{ReLU}(\text{FFN}_{\text{geo}}([\mathbf{h}_j \| \mathbf{h}_k])) \in \mathbb{R}^3, \tag{11}$$

where $[\mathbf{h}_j \| \mathbf{h}_k]$ denotes the concatenation of embeddings from joint $j$ and joint $k$. We apply an $\ell_2$ loss to the ground-truth offset vector $\mathbf{o}_{jk} = \mathbf{p}_k - \mathbf{p}_j$:

$$\mathcal{L}_{\text{geo}} = \sum_{(j,k) \in \mathcal{P}} \|\hat{\mathbf{o}}_{jk} - \mathbf{o}_{jk}\|_2^2, \tag{12}$$

where $\mathcal{P}$ is the set of joint pairs for which offset regression is performed.

- **Topological Task — LCA Prediction.** Given two nodes $(j_1, j_2)$, the model predicts their Least Common Ancestor (LCA) in the skeletal tree. Due to symmetry $\text{LCA}(j_1, j_2) = \text{LCA}(j_2, j_1)$, we first compute:

$$\mathbf{q}_{j_1 j_2} = \text{FFN}_{\text{query}}(\mathbf{h}_{j_1} + \mathbf{h}_{j_2}) \in \mathbb{R}^d, \tag{13}$$

$$\mathbf{k}_j = \text{FFN}_{\text{key}}(\mathbf{h}_j) \in \mathbb{R}^d, \quad \forall j \in \mathcal{J}. \tag{14}$$

We compute LCA probabilities using dot-product attention:

$$p_j = \frac{\exp(\mathbf{q}_{j_1 j_2}^\top \mathbf{k}_j)}{\sum_{j' \in \mathcal{J}} \exp(\mathbf{q}_{j_1 j_2}^\top \mathbf{k}_{j'})}, \tag{15}$$

and apply a cross-entropy loss with the ground-truth LCA label $j^\star$:

$$\mathcal{L}_{\text{lca}} = -\log p_{j^\star}. \tag{16}$$

- **Semantic Task — Contrastive Joint Name Matching.** We encourage node features to retain semantic consistency with their joint names. Let $\mathbf{h}_j \in \mathbb{R}^d$ be the node embedding, and let $\mathbf{e}_j^{\text{text}} \in \mathbb{R}^{512}$ be the CLIP-encoded joint name. We pass both through learnable projections:

$$\mathbf{z}_j^{\text{node}} = \text{FFN}_{\text{node}}(\mathbf{h}_j) \in \mathbb{R}^d, \tag{17}$$

$$\mathbf{z}_j^{\text{text}} = \text{FFN}_{\text{text}}(\text{CLIP}(j\text{-name})) \in \mathbb{R}^d. \tag{18}$$

Then we compute the InfoNCE loss across all joints in a batch:

$$\mathcal{L}_{\text{sem}} = -\sum_j \log \frac{\exp(\text{sim}(\mathbf{z}_j^{\text{node}}, \mathbf{z}_j^{\text{text}})/\tau)}{\sum_{j'} \exp(\text{sim}(\mathbf{z}_j^{\text{node}}, \mathbf{z}_{j'}^{\text{text}})/\tau)}, \tag{19}$$

where $\text{sim}(\cdot, \cdot)$ is cosine similarity and $\tau$ is the temperature.

**Final Loss.** The overall training loss is the weighted sum:

$$\mathcal{L}_{\text{graph}} = \lambda_{\text{geo}} \mathcal{L}_{\text{geo}} + \lambda_{\text{lca}} \mathcal{L}_{\text{lca}} + \lambda_{\text{sem}} \mathcal{L}_{\text{sem}}, \tag{20}$$

where $\lambda_{\text{geo}}, \lambda_{\text{lca}}, \lambda_{\text{sem}}$ are tunable hyperparameters.

### C.4 PROOF OF THEOREM

**Theorem.** *If a model can correctly determine the LCA for any pair of nodes $(i, j)$ in a tree, then the entire tree topology can be uniquely reconstructed.*

*Constructive proof.* We describe an explicit reconstruction procedure that uses only LCA queries.

**Step 1: Find the root.** Scan all nodes and pick the unique node $r$ such that for every other node $v$, $\text{LCA}(r, v) = r$. A non-root node $u$ fails this check because $\text{LCA}(u, \text{parent}(u)) = \text{parent}(u) \neq u$. Thus $r$ is identified.

**Step 2: Split into the root's child subtrees.** Consider all nodes except $r$. Place two nodes $u$ and $v$ into the same group iff $\text{LCA}(u, v) \neq r$. Nodes from different child subtrees of $r$ have LCA $= r$, so they fall into different groups; nodes from the same child subtree never produce $r$ as their LCA, so they fall into the same group. Hence each group is exactly one child subtree of $r$.

**Step 3: Identify each child of the root.** In each group $C$, find the unique node $c \in C$ such that for all $v \in C$, $\text{LCA}(c, v) = c$. This $c$ is the root of that group's subtree, i.e., a direct child of $r$. Add edge $(r, c)$.

**Step 4: Recurse.** Now treat each group $C$ as an independent problem: use the same two tests inside $C$ (with LCA restricted to pairs in $C$) to find the local root $c$ of $C$, split $C \setminus \{c\}$ by whether the LCA equals $c$, identify $c$'s children, add edges, and recurse until all groups are singletons.

This procedure terminates after assigning a unique parent to every non-root node, thereby reconstructing all edges. Because every step is determined solely by LCA answers and yields a unique outcome (unique root, unique grouping, unique local roots), the recovered tree is unique. $\square$

### C.5 RESIDUAL VQ-VAE

To convert continuous motion features into discrete tokens, we adopt a Residual Vector Quantized Variational Autoencoder (RVQVAE) as the tokenization backend. Compared to single-level VQ-VAE, the residual formulation improves expressiveness without increasing token sequence length, which is essential for high-fidelity motion reconstruction and efficient generation.

Given the encoder output $\mathbf{Z}_{\text{joint}} \in \mathbb{R}^{B \times T/r \times J \times W}$, we apply residual quantization in $R$ stages. Each stage learns a separate codebook $\mathcal{C}^{(r)} = \{\mathbf{c}_k^{(r)}\}_{k=1}^{K} \subset \mathbb{R}^d$, where $r = 1, \ldots, R$ and $K$ is the number of codewords.

The quantization is performed sequentially:

$$\mathbf{z}^{(0)} = \mathbf{Z}_{\text{joint}}, \mathbf{z}^{(r)} = \mathbf{z}^{(r-1)} - \text{Quantize}\left(\mathbf{z}^{(r-1)}; \mathcal{C}^{(r)}\right). \tag{21}$$

The final quantized feature is reconstructed as:

$$\hat{\mathbf{Z}}_{\text{joint}} = \sum_{r=1}^{R} \text{Quantize}\left(\mathbf{z}^{(r-1)}; \mathcal{C}^{(r)}\right). \tag{22}$$

We apply the same process to the global token $\mathbf{z}_{\text{token}}$ in parallel. Each quantized codeword index $z_{t,j}^{(r)} \in \{1, \ldots, K\}$ is stored as part of the discrete token matrix $\mathbf{Z} \in \mathbb{N}^{T/r \times R}$ for downstream modeling.

## D    EVALUATION METRICS

For **Text-to-Motion (T2M)**, follow existing works Guo et al. (2022b; 2023); Jiang et al. (2024); Wang et al. (2024); Zhu et al. (2025), we evaluate motion quality and text–motion alignment. Motion realism is measured by Fréchet Inception Distance (FID), while R-Precision (R@1/2/3) assess semantic consistency between motion and text.

Table 3: **Ablation Study on the OwO Design.**

| Method | MPJPE ↓ | | | MPJPE (no trans) ↓ | | | GeoDist ↓ | | |
|---|---|---|---|---|---|---|---|---|---|
| | H3D | Obj-XL | Zoo | H3D | Obj-XL | Zoo | H3D | Obj-XL | Zoo |
| Learnable Query | 0.1475 | 0.1585 | 0.1149 | 0.0751 | 0.1295 | 0.0737 | 5.04° | 19.44° | 16.37° |
| +Joint Name | 0.1759 | 0.1434 | 0.1324 | 0.0755 | 0.1005 | 0.0645 | 4.93° | 15.80° | 15.15° |
| **Ours** | **0.1084** | **0.0983** | **0.1008** | **0.0588** | **0.0787** | **0.0635** | **3.96°** | **12.12°** | **13.88°** |

**R-Precision.** R-Precision measures retrieval performance by computing the fraction of relevant items within the top-$R$ retrieved results. In text-to-motion, this means retrieving the correct motion from a database given a text query, or vice versa.

$$R\text{-Prec} = \frac{|\text{Rel} \cap \text{Top-}R|}{R} \qquad (23)$$

where Rel is the set of relevant items (ground-truth matches) and Top-$R$ is the set of retrieved items at rank $R$. The metric ranges from 0 to 1, with higher values indicating better retrieval accuracy.

**Fréchet Inception Distance (FID).** FID Guo et al. (2022a) measures the distributional distance between real and generated samples in a feature space, capturing both mean and covariance statistics. Lower FID indicates that generated samples are closer to real samples in distribution.

$$\text{FID} = \|\mu_r - \mu_g\|_2^2 + \text{Tr}\big(\Sigma_r + \Sigma_g - 2(\Sigma_r \Sigma_g)^{1/2}\big) \qquad (24)$$

where $\mu_r, \Sigma_r$ are the mean and covariance of real samples in the feature space, and $\mu_g, \Sigma_g$ are the corresponding statistics of generated samples. The first term measures the distance between means, while the second term accounts for differences in covariance structure.

**MPJPE.** Mean Per Joint Position Error (MPJPE) is a widely used metric to evaluate the accuracy of reconstructed 3D skeletons against ground-truth skeletons. It computes the average Euclidean distance between corresponding joints.

$$\text{MPJPE} = \frac{1}{T \cdot J} \sum_{t=1}^{T} \sum_{j=1}^{J} \|\hat{\mathbf{x}}_{t,j} - \mathbf{x}_{t,j}\|_2 \,, \qquad (25)$$

where $\hat{\mathbf{x}}_{t,j} \in \mathbb{R}^3$ denotes the predicted 3D position of joint $j$ at frame $t$, and $\mathbf{x}_{t,j} \in \mathbb{R}^3$ is the corresponding ground-truth joint position. $T$ is the number of frames and $J$ is the number of joints. The Euclidean norm $\|\cdot\|_2$ measures the spatial error per joint, and MPJPE averages this over all joints and frames.

**Geodesic Distance.** Geodesic distance is a metric to evaluate the difference between two 3D rotation matrices, often used for skeletal joint rotations. It measures the shortest distance along the manifold of the special orthogonal group $SO(3)$.

$$\text{Geo}(\hat{R}, R) = \arccos\left(\frac{\text{trace}(\hat{R}R^\top) - 1}{2}\right), \qquad (26)$$

where $\hat{R}, R \in SO(3)$ denote the predicted and ground-truth $3 \times 3$ rotation matrices, and $\text{trace}(\cdot)$ is the matrix trace operator. This formula computes the geodesic angular distance in radians between the two rotations. When averaged across joints and frames, this metric captures how well the predicted rotations align with the ground truth.

Table 4: **Ablation on RVQ Tokenizer Depth.**

| Method | MPJPE ↓ | | | MPJPE (no trans) ↓ | | | GeoDist ↓ | | |
|--------|------|--------|------|------|--------|------|------|--------|------|
| | H3D | Obj-XL | Zoo | H3D | Obj-XL | Zoo | H3D | Obj-XL | Zoo |
| RVQ-1 | 0.3663 | 0.1910 | 0.2200 | 0.1328 | 0.1439 | 0.0799 | 6.41° | 21.89° | 16.71° |
| RVQ-2 | 0.2232 | 0.1586 | 0.3503 | 0.1066 | 0.1237 | 0.0785 | 6.03° | 18.30° | 16.00° |
| RVQ-4 | 0.1052 | 0.1053 | 0.1157 | 0.0647 | 0.0869 | 0.0612 | 4.23° | 13.88° | 14.12° |
| RVQ-6 | 0.1084 | 0.0983 | 0.1008 | 0.0588 | 0.0787 | 0.0635 | 3.96° | 12.12° | 13.88° |
| RVQ-8 | 0.1229 | 0.0904 | 0.1526 | 0.0579 | 0.0684 | 0.0581 | 3.84° | 11.49° | 13.04° |

