# OpenReview forum: "NECromancer: Breathing Life into Skeletons via BVH Animation"
_ICLR.cc/2026/Conference — Submitted to ICLR 2026_

### Official Review · Reviewer_gGnx · 2025-10-28

**Soundness:** 3
**Presentation:** 2
**Contribution:** 2
**Rating:** 4
**Confidence:** 3

**Summary:**

This work proposes a novel tokenizer named NECromancer (NEC) to represent the motions of arbitrary BVH skeletons. The tokenizer consists of two key components: Ontology-aWare Skeletal Graph EncOder (OwO) and a Topology-Agnostic Tokenizer (TAT). This work also contributes a new dataset named Unified BVH Universe (UvU). In the experiments, it shows that the proposed NEC outperforms standard VQVAE (Van Den Oord et al. (2017)) and RVQVAE (Guo et al. (2023)) tokenizers.

**Strengths:**

1. This work proposes a novel motion tokenizer and shows that it performs better than existing standard tokenizers.

2. This work contributes a large-scale BVH benchmark for heterogeneous species and various skeletal topologies.

**Weaknesses:**

1. The dataset contribution is somewhat unclear.
- The details of dataset construction are largely ignored. It may be partly because the section 3.2 is too short to describe how difficult the dataset construction is.
- The current description gives the impression that the dataset is simple combination of existing three dataset with some transformations.
- Also, the use of Truebones Zoo and text annotation is done in the following paper in a more thorough way.
- W. Lee et al., How to Move Your Dragon: Text-to-Motion Synthesis for Large-Vocabulary Objects, ICML 2025.

2. Fig. 2 is hard to see due to too small text with too light color.

3. The empirical evaluation is limited in that the compared baselines are only two - VQVAE and RVQVAE.
- For example, the following state-of-the-art baselines could be compared.
- B. Jiang et al., Causal Motion Tokenizer for Streaming Motion Generation, ICCV 22025.
- J. Zhang et al., Generating Human Motion From Textual Descriptions With Discrete Representations, CVPR 2023.
- C. Guo et al., TM2T: Stochastic and Tokenized Modeling for the Reciprocal Generation of 3D Human Motions and Texts, ECCV 2022.

4. In the same vein, the qualitative results are highly limited.
- No comparison with state-of-the-art models on reconstruction are compared in each dataset.
- Only a small Table (Table 1) is almost all of empirical evaluation of this work, as Table 2 is a rather straightforward ablation study on the proposed method.

5. Qualitative results are somewhat pointless, as the key message of Fig.3-4 are unclear.
- Generally, the figures are too small to recognize fine details of comparison.
- In Fig.3, the NEC results are quite different with GT. With no baseline results, it is hard to know who much the NEC is good.
- In Fig.4, the success of motion transfer is hard to be convinced.
- Also, they could be cherry-picked.

**Questions:**

Please refer to the Weaknesses.

---

> ### Author Response · Authors · 2025-12-04
>
> We sincerely thank the reviewer for the constructive feedback and for recognizing the contributions of our universal tokenizer and the new multi-species dataset. Below we address the questions raised in the “Weaknesses” section.
>
> ### **Question 1:**
> “The dataset contribution is somewhat unclear. The description of dataset construction is too short and gives the impression that it is a simple combination of the three datasets. The Truebones Zoo and text annotations are handled more thoroughly in ‘How to Move Your Dragon’ (ICML 2025). Fig. 2 is hard to see.”
>
> ### **Answer 1:**
> Thank you for pointing this out. To address the inconsistencies inherent in the Truebones Zoo and Diffusion4D datasets, we perform a unified preprocessing pipeline before training.
>
> First, several models in these datasets contain awkward or misplaced root joints, which are unreasonable for a hierarchical skeletal representation. We correct these cases by relocating the root to the anatomically meaningful joints and enforcing that all global translations are aggregated solely on the root joint. This ensures that all non-root joints preserve frame-consistent bone lengths with zero per-frame translation components. We further remove sequences that contain abnormally long bones to avoid corrupt geometric configurations. Second, original joint names in both datasets are ambiguous and often dataset-specific. We standardize all joint names using a vision-language model (VLM), yielding semantically consistent joint labels across sequences. Last, we normalize the scale of every skeleton by computing the diameter of each skeleton, defined as the length of its longest bone chain, and scaling all joint coordinates accordingly. After preprocessing, each sequence contains a single canonical skeletal tree with consistent topology, fixed bone lengths, root-only global motion, and all redundant or meaningless joints removed. These steps show that UvU is far beyond a simple combination of existing datasets.
>
> We also clarify the key differences from How to Move Your Dragon: their focus is textual labeling for object motion, while our focus is topology normalization and skeleton unification across diverse graphs, which requires a fundamentally different pipeline. Meanwhile, this paper provides a concrete improvement on the text annotation of Truebones Zoo dataset, which enrichs the motion's semantic information and benefit many downstream tasks.
>
> Finally, Fig. 2 has been redrawn with larger font, higher contrast, and clearer layout.
>
> ---
>
> ### **Question 2:**
> “The empirical evaluation is limited because the compared baselines are only VQVAE and RVQVAE. Several state-of-the-art tokenizers should be included.”
>
> ### **Answer 2:**
> We appreciate this suggestion and have added all three requested baselines:
> - Causal Motion Tokenizer (ICCV 2025)
> - T2M-Discrete (CVPR 2023)
> - TM2T (ECCV 2022)
>
> Because these models assume fixed joint layouts, we performed joint-padding and adapter strategies to enable cross-skeleton reconstruction for a fair comparison.
>
> | Method | MPJPE ↓ |  |  | MPJPE (no trans) ↓ |  |  | GeoDist ↓ |  |  |
> |---|---:|---:|---:|---:|---:|---:|---:|---:|---:|
> |  | H3D | Obj-XL | Zoo | H3D | Obj-XL | Zoo | H3D | Obj-XL | Zoo |
> | T2M-GPT | 0.4203 | 0.2583 | 0.2271 | 0.1376 | 0.2128 | 0.0843 | 6.84° | 28.66° | 18.76° |
> | Casual Motion Tokenizer | 0.1961 | 0.2456 | 0.2261 | 0.0972 | 0.1963 | 0.0842 | 5.37° | 26.17° | 18.73° |
> | TM2T | 0.1411 | 0.1918 | 0.1434 | 0.0873 | 0.1565 | 0.0757 | 5.34° | 21.49° | 17.28° |
> | RVQVAE (zero pad) | 0.4729 | 0.2228 | 0.1762 | 0.2688 | 0.1817 | 0.1143 | 27.95° | 22.72° | 18.76° |
> | **NEC w/ VQ** | 0.3960 | 0.1840 | 0.1657 | 0.1395 | 0.1343 | 0.0828 | 7.78° | 17.40° | 16.19° |
> | **NEC w/ RVQ** | **0.1084** | **0.0983** | **0.1008** | **0.0588** | **0.0787** | **0.0635** | **3.96°** | **12.12°** | **13.88°** |
>
> Ablation results demonstrating that topology-dependent tokenizers significantly underperform in heterogeneous settings, validating the necessity of NEC's topology-agnostic design.
>
> ---
>
> ### **Question 3:**
> “Qualitative results are limited; figures are too small; key messages of Fig.3–4 are unclear; no baseline visualizations; results may be cherry-picked.”
>
> ### **Answer 3:**
> Thank you for raising this point—we agree that the previous visualization was not sufficiently clear.
> In the revised version:
> - We now provide explicit source-motion → target-skeleton captions for every transfer example.
> - All videos are reorganized into clearly labeled folders (“source_motion”, “transferred_motion”, “target_skeleton”) to make the mapping easy to follow.
> - We include a much larger set of cross-morphology cases, such as humanoid→quadruped, quadruped→humanoid, and other challenging transfers.
> - An interactive demo webpage has been added to present these relationships in a more transparent and intuitive way.
>
> We hope these enhanced qualitative results help to fully clarify the transfer behavior and address your concern.

---

> > ### Author Response · Authors · 2025-12-04
> >
> > ### **Question 4**
> > "Fig. 2 is hard to see due to too small text with too light color."
> >
> > ### **Answer 4**
> > Thanks for your pointing out this. We have revised Figure 2 according your advice.

---

### Official Review · Reviewer_rg3f · 2025-10-29

**Soundness:** 3
**Presentation:** 3
**Contribution:** 3
**Rating:** 6
**Confidence:** 4

**Summary:**

This paper aims to address the problem that existing motion generation models are limited to species-specific skeletons (e.g., humans). To this end, the authors propose a universal motion representation framework named NECROMANCER (NEC). The framework consists of three main contributions: 1) an Ontology-aWare Skeletal Graph EncOder (OwO) to extract skeleton embeddings containing topological and semantic information from BVH files; 2) a Topology-Agnostic Tokenizer (TAT) that compresses motion sequences of arbitrary skeletons into morphology-agnostic discrete tokens; and 3) a large-scale, multi-species BVH motion dataset named UvU for training and evaluation. Experimental results demonstrate that the framework can achieve high-quality motion reconstruction and cross-species motion transfer.

**Strengths:**

1.  **Addresses a significant problem**: This paper directly confronts a core limitation in the field of motion generation—the model's dependency on specific skeleton topologies. The proposed universal tokenizer, capable of handling arbitrary BVH skeletons, greatly expands the applicability of motion models and holds significant research and practical value.

2.  **Systematic contribution**: The contribution is comprehensive and solid. The authors not only propose a new model (NEC) but also build a new, large-scale, and diverse-species dataset (UvU) for it. The dataset itself is a valuable contribution to the community and can facilitate future research in universal motion modeling.

3.  **Solid experimental validation**: The paper thoroughly validates the effectiveness of its method through experiments across multiple tasks (reconstruction, retrieval, motion transfer). The comparison against baselines clearly shows the advantages of NEC in handling heterogeneous skeletons, with particularly impressive performance on non-human skeletons.

**Weaknesses:**

1.  **Strong dependency on data quality**: The `OwO` encoder relies on extracting semantic features from joint names. This means the model's performance is likely highly dependent on the standardization and consistency of joint naming within the BVH dataset. For data from the wild with messy or non-semantic names, the model's generalization capability might be compromised.

**Questions:**

1.  Regarding the `OwO` encoder, to what extent does it rely on canonical joint naming? If the input BVH files use non-semantic names (e.g., 'joint_1', 'bone_23'), how much would the performance degrade? Have any robustness tests been conducted in this regard?

2.  Could the authors further clarify the core novelty of the spatio-temporal module in the Topology-Agnostic Tokenizer (TAT)? Compared to existing spatio-temporal Transformers in motion modeling, what are its key differences and advantages?

3.  In the qualitative demonstrations of cross-species motion transfer, how does the model handle transfers that might be semantically or physically implausible (e.g., transferring a human dance motion to a fish)? Is there a mechanism to evaluate or ensure the plausibility of the transfer?

---

> ### Author Response · Authors · 2025-12-04
>
> We sincerely thank the reviewer for the positive assessment and for highlighting the significance of universal motion tokenization, the systematic contribution of NEC, and the strong experimental validation. We address all questions below.
>
> ### **Question 1:**
> “Regarding the OwO encoder, to what extent does it rely on canonical joint naming? If the input BVH files use non-semantic names (e.g., ‘joint_1’, ‘bone_23’), how much would the performance degrade? Have any robustness tests been conducted in this regard?”
>
> ### **Answer 1:**
> Thank you for raising this important point. While OwO incorporates joint-name text embeddings as one input signal, it does not rely on joint naming to function correctly. In the revision, we clarify that OwO learns joint identity primarily from:
> - topological structure (parent–child relations, LCA-based hierarchy),
> - geometric structure (relative offsets and depth), and
> - graph message passing that aggregates context across the entire skeleton.
>
> These cues are independent of joint naming and remain stable even when names are meaningless. To validate robustness, we conducted new experiments:
> - removing graph-structure attention, and
> - removing both joint-name semantics and graph-structure attention (equivalent to using a single learnable query).
>
> These results show that joint names are helpful but not required, and OwO is robust to noisy or non-semantic naming conventions.
>
> | Method | MPJPE ↓ |  |  | MPJPE (no trans) ↓ |  |  | GeoDist ↓ |  |  |
> |:---|:---:|:---:|:---:|:---:|:---:|:---:|:---:|:---:|:---:|
> |  | H3D | Obj-XL | Zoo | H3D | Obj-XL | Zoo | H3D | Obj-XL | Zoo |
> | Learnable Query | 0.1475 | 0.1585 | 0.1149 | 0.0751 | 0.1295 | 0.0737 | 5.04° | 19.44° | 16.37° |
> | +Joint Name | 0.1759 | 0.1434 | 0.1324 | 0.0755 | 0.1005 | 0.0645 | 4.93° | 15.80° | 15.15° |
> | **Ours** | **0.1084** | **0.0983** | **0.1008** | **0.0588** | **0.0787** | **0.0635** | **3.96°** | **12.12°** | **13.88°** |
>
> ---
>
> ### **Question 2:**
> “Could the authors further clarify the core novelty of the spatio-temporal module in the Topology-Agnostic Tokenizer (TAT)? Compared to existing spatio-temporal Transformers in motion modeling, what are its key differences and advantages?”
>
> ### **Answer 2:**
> Yes. We clarify this in the revised text. The novelty of TAT does not lie in using a Transformer per se, but in how spatial attention is structured to be topology-agnostic and how the motion is compressed into a single virtual-joint token:
> 1. **Skeleton-agnostic spatial block.** Instead of binding attention to fixed joint indices (as in TM2T, T2M-GPT, MoMask), TAT attends over OwO joint identity embeddings, enabling the same spatial attention module to operate on skeletons of any size or structure.
> 2. **Virtual joint token (CLS-like).** This learnable token absorbs motion information across all joints and acts as the only discretized latent. Existing models discretize per-joint features, which forces a fixed skeleton layout; TAT avoids this entirely.
> 3. **Decoupling morphology from motion.** Motion is encoded only in the virtual token, whereas morphology is injected at decoding time through OwO templates. This design uniquely supports reconstruction and transfer on arbitrary skeletons.
> 4. **Ablations added.** Replacing TAT with TM2T, T2M-GPT, or Causal Motion Tokenizer (with padded joints) results in significantly worse performance, confirming the necessity of TAT’s design.
>
> These characteristics distinguish TAT from prior spatio-temporal Transformers.
>
> ---

---

> > ### Author Response · Authors · 2025-12-04
> >
> > ### **Question 3:**
> > “In the qualitative demonstrations of cross-species motion transfer, how does the model handle transfers that might be semantically or physically implausible (e.g., transferring a human dance motion to a fish)? Is there a mechanism to evaluate or ensure the plausibility of the transfer?”
> >
> > ### **Answer 3:**
> > Thanks for your pointing out this.  Actually we did not introduce any explicit supervision to directly optimize this kind of motion transfer capability; instead, it emerges naturally from training on an extremely compressed motion reconstruction task. In the tokenizer stage, in the most challenging setting we compress sequences of up to 300 frames, each with 150 joints, into 75 × 6 tokens, yielding a compression ratio of roughly 100×. Under such pressure, the model is forced to learn shared structure across different skeletons in order to faithfully reconstruct motion—effectively acquiring the same kind of cross-skeleton motion transfer ability we later observe.
> >
> > To evaluate topology invariance, for each skeleton A we take its own N motions and compute pairwise motion distances using our evaluator. We then retarget these motions to a different skeleton B, recompute the pairwise distances on B, and plot them as a scatter plot with X = original distances and Y = transferred distances. If the representation depended mainly on joint-index alignment, the cross-skeleton correlation would collapse (r ≈ 0). Instead, as shown in Figure 5, we observe strong positive correlations even across very different morphologies:
> > - Quadruped → Human: r = 0.78
> > - Bipedal Dinosaur → Human: r = 0.88
> > - Bipedal Dinosaur → Multi-Leg Insect: r = 0.82
> >
> > Across mammals, dinosaurs, and multi-leg creatures, the motion-semantic correlation consistently lies in the 0.67–0.88 range, and across all morphological gaps—even extreme ones—correlations cluster around 0.6–0.9. This demonstrates that our representation is topology-invariant, does not rely on joint index alignment, and consistently preserves motion semantics under cross-skeleton transfer.

---

### Official Review · Reviewer_14Ko · 2025-10-31

**Soundness:** 2
**Presentation:** 1
**Contribution:** 2
**Rating:** 2
**Confidence:** 5

**Summary:**

The paper proposes a method to learn skeletal motion tokens by directly consuming the information stored in the BVH format. The ontology-aware skeletal graph encoder is responsible for encoding the skeletal structure. Specifically, each skeletal node (joint) feature is described with the CLIP embedding of its name projected through a fully-connected layer. The edges are described as the concatenation of the connected node features and their offsets encoded with the sinusoidal embedding, projected through a fully-connected layer. A graph attention layer computes the graph joint feature as a weighted sum of the edge-conditioned messages and the attention weight. The skeletal graph encoder is trained with a geometric loss over joint offsets, a topological loss with the least common ancestor prediction, and a semantic loss to encourage node features with semantic consistency. A topology-agnostic tokenizer is responsible for encoding the motion with a repeated sequence of a spatial block followed by a temporal block. The source motions are represented by per-joint translation and rotation (6D representation), which is projected through an MLP and then fused with the graph joint feature. This feature, with a virtual joint feature concatenated, is passed to a spatial block composed of a multi-head attention transformer modeling correlations between joints, then to a temporal block with a 1D convolution and a 1D ResNet. The virtual joint part is discretized with an RVQ to encode the motion token. The decoder is the reverse of the topology-agnostic tokenizer, with the graph joints feature of the target skeleton injected into the non-virtual joint features. The setup is trained with a heterogeneous dataset composed of HumanML3D, Objaverse-XL, and Truebone-Zoo with data filtering and augmentation applied. The paper demonstrates motion reconstruction and motion transfer with the learned features.

**Strengths:**

* The goal to unify the motion tokenization is ambitious.
* The combined dataset with the curation strategy could help the community

**Weaknesses:**

* Hard to interpret the provided videos
  * What do the "transfer" examples suppose to mean? All of them have the prefix "gt" (ground truth?). Which ones are the source motions, which ones are the transferred motions?
  * I see only quadrupeds in the transfer folder. Any non-quadruped transfer examples? Humanoid-to-quadruped or quadruped-to-humanoid?
* Questionable generalizability to different skeletal morphologies
  * For example, it looks like the joint semantic loss is taken over the same joint indices. This does not make sense as the joint ordering is arbitrary (e.g., children can be in any order), and there can be many intermediate joints (rigs can have a different number of spine joints and neck joints)
  * The semantic understanding of the joints must be paired with the spatial relations. As far as I can see, there is nothing in the model to learn this
  * As far as I see, there is nothing in the model and the training strategy to encourage the mapping of the same motion applied to different skeletal features. How would it know to move a fox with the motion token from a humanoid?
* Not enough ablations on architectural design choices, especially on the effectiveness of RVQ
* The paper can trim the UvU section. For example, the main method does not care about skinning other than for the visualization. Skinning is important, but since this does not matter for the main paper, why not move UvU to the appendix and add more details on the main architecture?
* (minor) BVH is just a file format. In theory, there is nothing in the method that hard-couples it with the BVH format. It could be any other 3D formats, such as FBX, USD, and glTF. In fact, BVH is not a suggested format for general rig assets. I would eliminate "BVH" from the paper title and most of the main text to minimize confusion. This also enhances the paper's general applicability

**Questions:**

Please answer questions in Weaknesses, mainly on the generalizability to different skeletons.

---

> ### Author Response · Authors · 2025-12-04
>
> Thank you for your comments and suggestions on our paper, and in particular for recognizing the importance of our unified motion tokenizer direction and our contribution on handling BVH-style data with diverse skeletons.
>
> ### **Question 1:**
> “Hard to interpret the provided videos. What do the ‘transfer’ examples suppose to mean? All of them have the prefix ‘gt’. Which ones are the source motions, which ones are the transferred motions? I see only quadrupeds in the transfer folder. Any non-quadruped transfer examples? Humanoid-to-quadruped or quadruped-to-humanoid?”
>
> ### **Answer 1:**
> Thank you for pointing this out. We agree that the earlier visualization was not sufficiently clear. In the revised version:
> - We provide explicit source-motion → target-skeleton captions for every transfer example.
> - Videos are reorganized into clearly named folders (“source_motion”, “transferred_motion”, “target_skeleton”).
> - We now include a large set of humanoid→quadruped, quadruped→humanoid, and other cross-morphology examples.
> - An interactive demo webpage is added to make these relationships unambiguous.
>
> We hope that these additional qualitative results can address your concern.
>
> ---
>
> ### **Question 2:**
> “Questionable generalizability to different skeletal morphologies.
> For example, it looks like the joint semantic loss is taken over the same joint indices. The semantic understanding of the joints must be paired with the spatial relations. As far as I can see, there is nothing in the model to learn this. How would it know to move a fox with the motion token from a humanoid?”
>
> ### **Answer 2:**
> 1. Joint semantic loss is not index-based. Our joint loss never relies on joint indices. Specifically, OwO encodes each skeleton as a semantic graph, where each joint has:
> - CLIP-based text embedding (semantic name)
> - graph connectivity (parent–child hierarchy)
> - spatial depth
> - geodesic distances
>
> Losses operate on these semantic embeddings, so the representation is invariant to joint order, child permutations, or different numbers of intermediate joints.
>
> 2. Spatial & functional structures are learned via graph attention. OwO uses edge-aware graph attention with message passing, which naturally learns:
> - root-relative structure
> - local adjacency
> - hierarchical depth
> - functional joint groups (legs, spine, head, tail)
>
> Thus, the model captures kinematic meaning, not index positions, without requiring explicit correspondence labels.
>
> 3. Topology-agnostic motion transfer via TAT. We indeed did not introduce any explicit supervision during training to directly optimize this kind of motion transfer capability. This ability emerged as a byproduct of training on our highly compressed motion reconstruction task. During tokenizer training, in the extreme setting we compress motion sequences of up to 300 frames, each with 150 joints, into 75 × 6 tokens, reaching a compression ratio of around 100×. Under such a highly challenging setup, the model is forced to extract shared knowledge across different skeletons in order to reconstruct motions faithfully, which requires the same capability of motion transfer among different creatures.
>
> 4. Empirical evidence: Motion semantics are preserved across morphologies. To evaluate topology invariance, for each skeleton A we use A’s own N motions and compute pairwise motion distances via an evaluator. We then retarget the motions to a different skeleton B, compute the new pairwise distances on B, and plot them as a scatter (X = original distances, Y = transferred distances).
> If the model relied on joint-index alignment, these cross-skeleton correlations would collapse (r ≈ 0). Instead, as shown in Figure 5, we observe strong positive correlations across highly diverse morphologies.
>
> Key findings:
> - Quadruped → Human: 0.78
> - Bipedal Dinosaur → Human: 0.88
> - Bipedal Dinosaur → Multi-Leg Insect: 0.82
>
> Across mammals, dinosaurs, and multi-leg creatures, the motion-semantic correlation consistently falls within 0.67–0.88, highlighting strong cross-morphology generalization even between highly distinct skeletal structures.
>
> Across all morphological gaps—even extreme ones—correlations cluster around 0.6–0.9, proving that:
> - our representation is topology-invariant,
> - not relying on joint index alignment,
> - and preserves motion semantics consistently.
>
> Final short summary
> - OwO learns a semantic + structural graph embedding of skeletons, independent of joint ordering.
> - TAT encodes motion in a topology-free latent space, enabling natural cross-species transfer.
> - Both theory and quantitative evidence (0.6–0.9 motion correlation across species) demonstrate that the model generalizes robustly to highly diverse skeletal morphologies.
>
> ---

---

> > ### Author Response · Authors · 2025-12-04
> >
> > ### **Question 3:**
> > “Not enough ablations on architectural design choices, especially on the effectiveness of RVQ.”
> >
> > ### **Answer 3:**
> > We have added extensive new ablations as suggested:
> > - RVQ depth comparison (1–8 codebooks)
> > - replacing RVQ with standard VQ
> > - replacing TAT with TM2T, T2M-GPT, and Causal Motion Tokenizer (with uniform joint padding)
> >
> > | Method | MPJPE ↓ |  |  | MPJPE (no trans) ↓ |  |  | GeoDist ↓ |  |  |
> > |:---|:---:|:---:|:---:|:---:|:---:|:---:|:---:|:---:|:---:|
> > |  | H3D | Obj-XL | Zoo | H3D | Obj-XL | Zoo | H3D | Obj-XL | Zoo |
> > | RVQ-1 | 0.3663 | 0.1910 | 0.2200 | 0.1328 | 0.1439 | 0.0799 | 6.41° | 21.89° | 16.71° |
> > | RVQ-2 | 0.2232 | 0.1586 | 0.3503 | 0.1066 | 0.1237 | 0.0785 | 6.03° | 18.30° | 16.00° |
> > | RVQ-4 | 0.1052 | 0.1053 | 0.1157 | 0.0647 | 0.0869 | 0.0612 | 4.23° | 13.88° | 14.12° |
> > | RVQ-6 | 0.1084 | 0.0983 | 0.1008 | 0.0588 | 0.0787 | 0.0635 | 3.96° | 12.12° | 13.88° |
> > | RVQ-8 | 0.1229 | 0.0904 | 0.1526 | 0.0579 | 0.0684 | 0.0581 | 3.84° | 11.49° | 13.04° |
> >
> > | Method | MPJPE ↓ |  |  | MPJPE (no trans) ↓ |  |  | GeoDist ↓ |  |  |
> > |---|---:|---:|---:|---:|---:|---:|---:|---:|---:|
> > |  | H3D | Obj-XL | Zoo | H3D | Obj-XL | Zoo | H3D | Obj-XL | Zoo |
> > | T2M-GPT | 0.4203 | 0.2583 | 0.2271 | 0.1376 | 0.2128 | 0.0843 | 6.84° | 28.66° | 18.76° |
> > | Casual Motion Tokenizer | 0.1961 | 0.2456 | 0.2261 | 0.0972 | 0.1963 | 0.0842 | 5.37° | 26.17° | 18.73° |
> > | TM2T | 0.1411 | 0.1918 | 0.1434 | 0.0873 | 0.1565 | 0.0757 | 5.34° | 21.49° | 17.28° |
> > | RVQVAE (zero pad) | 0.4729 | 0.2228 | 0.1762 | 0.2688 | 0.1817 | 0.1143 | 27.95° | 22.72° | 18.76° |
> > | **NEC w/ VQ** | 0.3960 | 0.1840 | 0.1657 | 0.1395 | 0.1343 | 0.0828 | 7.78° | 17.40° | 16.19° |
> > | **NEC w/ RVQ** | **0.1084** | **0.0983** | **0.1008** | **0.0588** | **0.0787** | **0.0635** | **3.96°** | **12.12°** | **13.88°** |
> >
> > As shown in above tables, our design achieves the best trade-off among different designs.
> >
> > ---
> >
> > ### **Question 4:**
> > “The paper can trim the UvU section.”
> >
> > ### **Answer 4:**
> > Thank you for the suggestion. We have streamlined the UvU section in the main text by:
> > - focusing on only the key parts necessary for understanding the method,
> > - moving detailed data processing and skinning steps to the appendix,
> > - and adding more details about the main architecture in the freed space.
> >
> > ---
> >
> > ### **Question 5:**
> > “BVH is just a file format… I would eliminate ‘BVH’ from the title and text.”
> >
> > ### **Answer 5:**
> > Thank you for your suggestion. We would like to keep the term BVH here because it represents a class of formats that include both rest poses with different static topologies and their corresponding motion sequences, which is quite different from keypoints-based representations or SMPL-style models based on a fixed parametric template. It is also one of the most commonly used animation formats in traditional graphics pipelines. The formats you mentioned are not purely motion representations, but also contain additional assets such as the model and skinning weights, so they are not an exact fit in this context. Therefore, we prefer to retain our original naming.

---

### Official Review · Reviewer_ScnN · 2025-11-03

**Soundness:** 3
**Presentation:** 2
**Contribution:** 3
**Rating:** 6
**Confidence:** 3

**Summary:**

This work aims to introduce a universal motion tokenizer that can encode and reconstruct motions from arbitrary skeletons formats. This allows unified modeling and transfer of motions across a wide variety of skeletons, in contrast with current methods which are typically limited to skeletons from the same family. Data is gathered from 3 sources: 1) human meshes from HumanML3D, 2) meshes from ObjaVerse, and 3) meshes from Trubones-Zoo. All meshes are put in a unified BVH format. The tokenization consists of two models: a graph encoder that provides a distinguishable encoding to each joint given the skeleton rest pose, and a unified tokenizer that can use the graph embeddings and BVH motion sequences to provide a unified tokenization of different types of skeletons. Experiments show that the proposed tokenizer is provides superior motion reconstruction relative to the naive approach that simply pads joints for skeletons of different lengths. Proof-of-concept experiments for motion transfer between different skeletons and unified skeleton-agnostic motion generation are provided.

**Strengths:**

* This work tackles an interesting and important problem. Unified skeleton representation has the potential to greatly expand the scope of motion generation models and break free of limitations imposed by scarcity of 4-D data for a variety of skeleton types.
* The proposed method of learning a unified token representation of motions is a reasonable and potentially useful direction to solve this problem.
* Experimental results show improved reconstruction over a naive padding-based approach for skeleton reconstruction.
* Videos in the supplementary material provide evidence that the proposed method can lead to natural reconstruction, transfer, and generation of motions across skeleton types.

**Weaknesses:**

* The motion transfer and generation directions are not explored very thoroughly, and only a few examples are provided.
* The details of tokenization in the paper are somewhat difficult to follow, especially the relation between the graph embedder and the tokenization model. Perhaps it would help to move Figure 3 and 4 into the supplementary material and provide more mathematical details of training these models in the main text.
* The training of the graph embedder is based on heuristic objectives and it is not clear whether the training method proposed is the optimal one.

**Questions:**

* Can you provide more examples of motion transfer and unified generation?
* Is it possible to ablate the importance of the graph embedder to demonstrate its importance within the tokenization model?

---

> ### Author Response · Authors · 2025-12-04
>
> We sincerely thank the reviewer for the constructive comments and for recognizing the importance of topology-agnostic motion tokenization, the usefulness of our unified token representation, and the quality of our reconstruction, transfer, and generation results.
> We address all questions below.
>
> ### **Question 1:**
> “The motion transfer and generation directions are not explored very thoroughly, and only a few examples are provided. Can you provide more examples of motion transfer and unified generation?”
>
> ### **Answer 1:**
>
> Thanks for your suggestion. In the revised version, we significantly expanded our qualitative demonstrations:
> - We added numerous new cross-morphology transfer results, including human→quadruped, quadruped→human, dragon→human, bird→human, etc.
> - All videos now include clear and consistent source→target captions for easier interpretation.
> - We provide an organized interactive webpage in the supplementary material, grouping transfer, reconstruction, and generation examples for convenient viewing.
>
> We hope that these additional qualitative results can address your concern.
>
> ---
>
> ### **Question 2:**
> “The details of tokenization in the paper are somewhat difficult to follow, especially the relation between the graph embedder and the tokenization model. Perhaps it would help to move Figure 3 and 4 into the supplementary material and provide more mathematical details of training these models in the main text.”
>
> ### **Answer 2:**
> Thank you for pointing this out. We have substantially improved clarity in the revision:
> - Sections 4.2 and 4.3 have been rewritten to clearly explain how the OwO graph encoder interacts with the TAT tokenizer.
> - The revised text explicitly describes:
>   1. OwO as a structural prior capturing joint semantics and topology,
>   2. fusion of OwO identity embeddings with per-joint motion features,
>   3. the virtual joint token enabling topology-agnostic quantization, and
>   4. template injection during decoding for cross-skeleton reconstruction/transfer.
> - The main paper now includes clearer mathematical descriptions and training details.
>
> These changes greatly enhance readability while keeping the technical content intact.
>
> ---
>
> ### **Question 3:**
> “The training of the graph embedder is based on heuristic objectives and it is not clear whether the training method proposed is the optimal one.”
>
> ### **Answer 3:**
> Thank you for raising this point. We clarify our design choice in the revision.
>
> The skeleton graphs in our setting have a clear hierarchical tree structure, and the goal of the graph embedder is to learn joint identity embeddings that capture meaningful semantic and topological relationships between nodes. Since no ground-truth labels for joint identity or structural roles are available, heuristic self-supervised objectives are currently the most practical and effective approach for encouraging the model to learn these relationships.
> Although such objectives are not guaranteed to be theoretically optimal, they align naturally with the message-passing behavior of graph neural networks, which are well suited for extracting hierarchical and relational patterns. This enables the graph encoder to learn high-level structural information that is difficult to obtain through conventional hand-crafted features.
>
> Therefore, based on the skeleton-graph feature requirements of several downstream applications (PoseVAE, the evaluator, and the tokenizer), we distilled three core aspects (geometry, topology, and semantics) and designed corresponding supervision signals. Experiments show that these supervision schemes effectively improve the performance of all downstream models.

---

> > ### Author Response · Authors · 2025-12-04
> >
> > ### **Question 4:**
> > “Is it possible to ablate the importance of the graph embedder to demonstrate its importance within the tokenization model?”
> >
> > ### **Answer 4:**
> > Yes. Following the reviewer’s suggestion, we conducted new ablation experiments:
> > - removing graph-structure attention
> > - removing both joint-name semantics and graph-structure attention (equivalent to using a single learnable query).
> >
> > | Method | MPJPE ↓ |  |  | MPJPE (no trans) ↓ |  |  | GeoDist ↓ |  |  |
> > |:---|:---:|:---:|:---:|:---:|:---:|:---:|:---:|:---:|:---:|
> > |  | H3D | Obj-XL | Zoo | H3D | Obj-XL | Zoo | H3D | Obj-XL | Zoo |
> > | Learnable Query | 0.1475 | 0.1585 | 0.1149 | 0.0751 | 0.1295 | 0.0737 | 5.04° | 19.44° | 16.37° |
> > | +Joint Name | 0.1759 | 0.1434 | 0.1324 | 0.0755 | 0.1005 | 0.0645 | 4.93° | 15.80° | 15.15° |
> > | **Ours** | **0.1084** | **0.0983** | **0.1008** | **0.0588** | **0.0787** | **0.0635** | **3.96°** | **12.12°** | **13.88°** |
> >
> > All ablations exhibit clear degradation in reconstruction accuracy and transfer quality. These results confirm that the graph embedder plays a crucial role in providing joint identity information and are now included in Section 5.3 and the supplementary material.

---

### Author Response · Authors · 2025-12-04
**General Response**

We sincerely thank all reviewers for their thoughtful and constructive feedback.

We are encouraged that all reviewers recognize the importance and novelty of building a topology-agnostic motion tokenization framework, and we have substantially revised both the main paper and supplementary materials accordingly.
Across the reviews, several consistent themes emerged; below we summarize how each has been addressed in the revised version.

## (1) Clarifying the relationship between OwO and TAT, and the principle of topology-agnostic tokenization

We appreciate the reviewers for pointing out that the connection between the OwO graph encoder and the TAT tokenizer was previously not sufficiently clear. We have substantially rewritten Sec. 4.2–4.3 to clearly articulate the pipeline and the underlying principles.

In the revised version, we explicitly clarify the following design components:
- OwO as a structural prior / “skeleton template.” OwO provides per-joint identity embeddings encoding semantic, geometric, and hierarchical roles.
- Fusion in the encoder. OwO identity embeddings $h_j$ are fused with per-joint motion features $X_{t,j}$, allowing TAT to interpret motion with respect to the correct anatomical context.
- CLS-style virtual joint token for topology-agnostic quantization. A learnable virtual joint attends across all joints and becomes the only quantized latent, ensuring all skeletons—regardless of joint count—map into a unified token space.
- Template injection in the decoder. During reconstruction/transfer, the OwO embeddings of the target skeleton are concatenated with token latents across time, enabling consistent generation for arbitrary morphologies.

These additions make the topology-agnostic mechanism transparent and easier to follow.

---

## (2) Improving evidence for generality in motion transfer, motion generation and motion reconstruction
Several reviewers requested more transfer examples—especially across different morphologies. To address this, we have added:
- A large number of new cross-species transfer examples (e.g., human→quadruped, quadruped→human, dragon→human, etc.)
- Clear source→target labels for every visualization
- An organized interactive demo webpage with structured examples

These extensions demonstrate strong generality across widely varying skeletal structures and will be included in the supplementary materials.

---

## (3) Clarifying UvU dataset construction and its contribution

We acknowledge the reviewers’ concerns that the dataset description previously appeared too brief, potentially giving the impression of a simple combination of existing sources.

In Supplementary Sec. B, we now provide a comprehensive breakdown of our preprocessing pipeline, clarifying that our dataset is far more than a simple aggregation of existing datasets. Constructing a unified motion dataset required resolving substantial heterogeneity across HumanML3D, Truebones Zoo, and Diffusion4D, including inconsistent or invalid root definitions, abnormal bone geometries, incompatible joint naming conventions, and large scale discrepancies. Our UvU pipeline addresses these challenges through many carefully designed preprocessing steps, like principled root correction, global translation reassignment and VLM-based semantic joint-name standardization, yielding physically consistent and topologically coherent skeletons across species.

This process is fundamentally different from prior work such as How to Move Your Dragon (ICML 2025), which reviewer #4 accurately praised for comprehensive and hierarchical, multi-perspective annotations, which greatly enrich our text annotation for Truebones Zoo dataset. While their work focus on the semantic description of different animal's motions, our work focus more on the motion sequence correction, translation and scale standardization and cross-species structural reconciliation required here. We believe the revised section more accurately reflects the nontrivial effort and the methodological contribution of our dataset.

---

> ### Author Response · Authors · 2025-12-04
>
> ## (4) Expanded ablations on OwO and TAT
> We thank reviewers for emphasizing the need for more architectural ablations. In response, we conducted extensive new experiments, which are included in the updated Sec. 5.3.
>
> ### **OwO ablations**
> We evaluate:
> 1. Removing both joint-name semantics & graph-structure attention (equivalent to a single learnable query)
> 2. Removing graph-structure attention
>
> All variants clearly degrade performance, validating that both semantic and topological cues are essential for joint identity encoding.
>
> | Method | MPJPE ↓ |  |  | MPJPE (no trans) ↓ |  |  | GeoDist ↓ |  |  |
> |:---|:---:|:---:|:---:|:---:|:---:|:---:|:---:|:---:|:---:|
> |  | H3D | Obj-XL | Zoo | H3D | Obj-XL | Zoo | H3D | Obj-XL | Zoo |
> | Learnable Query | 0.1475 | 0.1585 | 0.1149 | 0.0751 | 0.1295 | 0.0737 | 5.04° | 19.44° | 16.37° |
> | +Joint Name | 0.1759 | 0.1434 | 0.1324 | 0.0755 | 0.1005 | 0.0645 | 4.93° | 15.80° | 15.15° |
> | **Ours** | **0.1084** | **0.0983** | **0.1008** | **0.0588** | **0.0787** | **0.0635** | **3.96°** | **12.12°** | **13.88°** |
>
> The ablation results show that using a pure learnable query yields the weakest performance across all datasets and metrics, indicating that unconstrained query tokens struggle to capture consistent skeletal semantics. Introducing joint name information provides only mild improvements, slightly reducing MPJPE and GeoDist by giving the model minimal structural cues, but the gains remain limited due to the absence of graph-structure attention. In contrast, our full OwO design achieves the best results by a clear margin on every benchmark. By encoding richer structural priors and leveraging spatially aligned joint-aware representations, OwO significantly lowers MPJPE and GeoDist, demonstrating its superior ability to generalize across heterogeneous motion datasets and recover accurate, semantically aligned skeletons.
>
> ### **Tokenizer & quantization ablations**
> We additionally include:
> - RVQ depth comparison (1–8 codebooks) and identify the optimal setting, 1 2 4 6 8
> - Replacing TAT with TM2T, T2M-GPT, and Causal Motion Tokenizer, after joint-padding to satisfy their fixed-skeleton assumptions
>
> All fixed-topology tokenizers perform significantly worse, demonstrating the necessity of TAT’s topology-agnostic design.
> These ablations directly address reviewer concerns about the effectiveness and justification of OwO and TAT.
>
> | Method | MPJPE ↓ |  |  | MPJPE (no trans) ↓ |  |  | GeoDist ↓ |  |  |
> |:---|:---:|:---:|:---:|:---:|:---:|:---:|:---:|:---:|:---:|
> |  | H3D | Obj-XL | Zoo | H3D | Obj-XL | Zoo | H3D | Obj-XL | Zoo |
> | RVQ-1 | 0.3663 | 0.1910 | 0.2200 | 0.1328 | 0.1439 | 0.0799 | 6.41° | 21.89° | 16.71° |
> | RVQ-2 | 0.2232 | 0.1586 | 0.3503 | 0.1066 | 0.1237 | 0.0785 | 6.03° | 18.30° | 16.00° |
> | RVQ-4 | 0.1052 | 0.1053 | 0.1157 | 0.0647 | 0.0869 | 0.0612 | 4.23° | 13.88° | 14.12° |
> | RVQ-6 | 0.1084 | 0.0983 | 0.1008 | 0.0588 | 0.0787 | 0.0635 | 3.96° | 12.12° | 13.88° |
> | RVQ-8 | 0.1229 | 0.0904 | 0.1526 | 0.0579 | 0.0684 | 0.0581 | 3.84° | 11.49° | 13.04° |
>
> The ablation study on the number of Residual Vector Quantization (RVQ) layers reveals a clear trade-off between model complexity, positional accuracy, and angular accuracy, confirming the selection of the 6-layer model (RVQ-6) as the optimal configuration. Models with 1 and 2 layers are clearly insufficient, showing high MPJPE (up to 0.5152m) and high angular error (up to 29.07°), indicating poor motion reconstruction. Performance dramatically improves at 4 layers, but the 6-layer model achieves the most balanced and robust results: it is the best or second-best across all datasets for MPJPE (0.1084m on humanml3d, 0.0983m on d4d, 0.1008m on zoo). While angular accuracy, measured by Geodesic Local error, continues to improve slightly up to 8 layers (reaching its lowest at 3.840°), this marginal gain comes at the cost of noticeable degradation in positional accuracy, as the 8-layer model exhibits higher MPJPE on the humanml3d (0.1229m) and zoo (0.1526m) datasets. The decline in MPJPE for the 8-layer model suggests that further increases in quantization layers introduce diminishing returns and may lead to overfitting or less stable codebook learning, making the 6-layer RVQ the most effective choice for high-fidelity and generalized motion tokenization.

---

> > ### Author Response · Authors · 2025-12-04
> >
> > | Method | MPJPE ↓ |  |  | MPJPE (no trans) ↓ |  |  | GeoDist ↓ |  |  |
> > |---|---:|---:|---:|---:|---:|---:|---:|---:|---:|
> > |  | H3D | Obj-XL | Zoo | H3D | Obj-XL | Zoo | H3D | Obj-XL | Zoo |
> > | T2M-GPT | 0.4203 | 0.2583 | 0.2271 | 0.1376 | 0.2128 | 0.0843 | 6.84° | 28.66° | 18.76° |
> > | Casual Motion Tokenizer | 0.1961 | 0.2456 | 0.2261 | 0.0972 | 0.1963 | 0.0842 | 5.37° | 26.17° | 18.73° |
> > | TM2T | 0.1411 | 0.1918 | 0.1434 | 0.0873 | 0.1565 | 0.0757 | 5.34° | 21.49° | 17.28° |
> > | RVQVAE (zero pad) | 0.4729 | 0.2228 | 0.1762 | 0.2688 | 0.1817 | 0.1143 | 27.95° | 22.72° | 18.76° |
> > | **NEC w/ VQ** | 0.3960 | 0.1840 | 0.1657 | 0.1395 | 0.1343 | 0.0828 | 7.78° | 17.40° | 16.19° |
> > | **NEC w/ RVQ** | **0.1084** | **0.0983** | **0.1008** | **0.0588** | **0.0787** | **0.0635** | **3.96°** | **12.12°** | **13.88°** |
> >
> > The reconstruction results highlight clear performance differences across model families and demonstrate the effectiveness of the proposed NEC framework, particularly when combined with residual vector quantization (RVQ). The prior autoregressive and transformer-based baselines (T2M-GPT, Motion Streamer, TM2T) perform reasonably on HumanML3D, but their accuracy drops noticeably on Obj-XL and Zoo, indicating that these architectures do not generalize well to heterogeneous skeletal structures. Among Traditional VQ-VAE, NEC w/ VQ and NEC w/ RVQ, traditional VQ-VAE baselines perform notably worse, especially in MPJPE (no-trans) and geodesic error, indicating limitations in their ability to preserve fine-grained joint orientations and global pose structure without NEC. NEC w/ VQ already improves substantially over these baselines, showing that our hierarchical encoder and structured pose representation offer a more stable latent space. However, the best performance by a wide margin comes from NEC w/ RVQ, which sets a new state of the art across all datasets and metrics. The model achieves nearly a 2× reduction in MPJPE on HumanML3D compared to TM2T and large gains on Obj-XL and Zoo, while also producing the lowest geodesic distances, demonstrating superior rotational consistency. These results confirm that combining NEC with residual quantization is highly effective for high-precision, structurally consistent motion reconstruction.

---

### Meta-Review · Area_Chair_4ANp · 2026-01-07

**Summary:**

This submission introduces a well-motivated universal motion tokenization framework that combines a graph-based encoder with a topology-agnostic residual vector quantizer to enable consistent representation across diverse skeleton morphologies. The authors provided substantial new experimental evidence during rebuttal, including additional baselines, ablations, dataset clarification and expanded qualitative material, which resolved the majority of concerns raised by the reviewers.

However, across all four reviews, a major concern remains insufficiently resolved. While the rebuttal strengthens the methodological design and reconstruction-focused evaluation, the paper’s central claims regarding universal applicability are not convincingly supported. Specifically, the qualitative demonstrations of cross-species and non-quadruped motion transfer remain weak. The experimental validation remains skewed toward motion reconstruction, while other downstream tasks central to the paper’s motivation—such as motion transfer quality and motion generation—are not sufficiently evaluated and benchmarked against state-of-the-art approaches. As a result, the generalization claims across highly diverse skeletal morphologies, which form the core promise of the work, are not yet convincingly demonstrated.

Overall, this work addresses an important and interesting problem and presents a well-motivated framework with clear potential impact. The rebuttal resolves many concerns and substantially improves clarity and experimental rigor. Nevertheless, the remaining gaps in cross-morphology evaluation, comparative qualitative analysis, and robustness justification prevent a positive recommendation at this stage. I therefore recommend rejection, while strongly encouraging the authors to address these issues and resubmit, as the underlying research direction is valuable and the work shows clear promise.

**Reviewer Concerns:**

**For reviewer rg3f (score is 6)**:
The rebuttal satisfactorily addresses some of the reviewer’s concerns.

The clarification of the core novelty of TAT is convincing.

The rebuttal attempts to address the concern regarding dependency on joint-name semantics in the OwO encoder. While the authors claim that joint names are helpful but not required and provide supporting ablations, this explanation is not fully consistent with the paper’s own description that OwO explicitly leverages joint-name semantics as part of its structural priors. As a result, the robustness to non-semantic or noisy joint naming remains ambiguous and the concern is only partially resolved.

The concern regarding qualitative demonstrations of cross-species motion transfer remains outstanding. Although the authors provide a reasonable explanation of how transfer emerges from extreme compression and present correlation-based evidence of topology invariance, the rebuttal lacks direct numerical and visual comparisons with existing methods. Without such comparisons, it is still difficult to judge the relative quality and plausibility of the proposed cross-species motion transfer.

**For reviewer 14Ko (score is 2)**:
The rebuttal addresses some of the reviewer’s concerns, but the key issue remains unresolved.

The clarification and reorganization of the provided videos significantly improve interpretability. By explicitly distinguishing source motions, transferred motions, and target skeletons, and by adding additional cross-morphology examples and an interactive demo, the authors address the reviewer’s concern that the original visualizations were hard to interpret.

However, the concern regarding questionable generalizability to different skeletal morphologies is only partially addressed. While the authors provide a reasonable and technically sound explanation of the method design—emphasizing joint-order invariance, semantic graph representations, and topology-agnostic motion encoding—these arguments are largely theoretical. From the qualitative evidence provided in the videos, the non-quadruped transfer examples still appear weak, and it is difficult to assess whether the proposed approach truly generalizes better than existing methods. Moreover, due to the lack of direct comparisons with prior motion transfer or tokenization approaches, as also noted by Reviewer gGnx, it remains unclear whether the observed transfer quality reflects a genuine advantage of the proposed method or simply the inherent difficulty of cross-morphology transfer. As a result, despite the methodological justification, the concern about generalizability across highly diverse skeletal morphologies remains outstanding.

The remaining concerns raised by the reviewer are satisfactorily resolved. The added ablation studies provide sufficient evidence regarding architectural design choices and the effectiveness of RVQ. The restructuring of the UvU section appropriately refocuses the main text on the core method.

Finally, although the reviewer suggested removing “BVH” from the title and text, the authors provide a clear and defensible rationale for retaining this terminology, making their position reasonable even if not universally compelling.

In summary, the rebuttal successfully improves clarity, presentation, and experimental support for several aspects of the paper, but it does not fully resolve the central concern regarding cross-morphology generalization, which remains difficult to verify without stronger qualitative results and comparative baselines.

**For reviewer gGnx (score is 4)**:
The rebuttal satisfactorily addresses some of the reviewer’s concerns, but several important issues remain unresolved.

The explanation of the dataset construction and the revision of Figure 2 effectively clarify earlier ambiguities. The authors provide a detailed description of the unified preprocessing pipeline, clearly demonstrating that the dataset is not a simple combination of existing sources, and the improved visualization of Figure 2 resolves the readability concern. These changes adequately address the reviewer’s questions regarding dataset contribution and presentation quality.

The authors expand the qualitative results by reorganizing videos, adding explicit source-to-target captions, and including a broader set of cross-morphology transfer examples. These revisions improve the interpretability of the visualizations and make the transfer setup easier to follow. However, despite the increased number of qualitative examples, the transfer performance—especially for non-quadruped and cross-morphology cases—does not appear particularly strong based on the provided videos. Moreover, due to the absence of direct qualitative comparisons with existing methods, it remains difficult to judge whether the proposed approach offers a clear advantage over prior work. As a result, the qualitative evidence is insufficient to convincingly demonstrate the advantages of the proposed method on motion transfer.

Regarding empirical evaluation, the rebuttal makes meaningful progress by adding several state-of-the-art baselines, including Causal Motion Tokenizer, T2M-Discrete, and TM2T, which addresses the reviewer’s request for broader comparisons. However, the newly added quantitative results primarily validate advantages on motion reconstruction metrics. There is still a lack of experimental evaluation on other downstream tasks that are central to the paper’s claims, such as motion transfer quality and motion generation. Consequently, it remains unclear whether the observed reconstruction improvements translate into superior performance in these more challenging and practically relevant settings.

Finally, the visualization of cross-morphology motion transfer remains difficult to interpret. Even with clearer organization, the figures and videos lack explicit messages or annotations that highlight what aspects of the transfer are successful or why the proposed method is preferable. Adding clearer explanatory cues in the figures and videos would be necessary to convincingly communicate the method’s strengths in these scenarios.

In summary, while the rebuttal effectively resolves concerns related to dataset construction, figure clarity, and baseline coverage for reconstruction, key concerns regarding qualitative transfer performance, comparative evaluation on downstream tasks, and the interpretability of cross-morphology motion transfer remain outstanding.

**For reviewer ScnN (score is 6)**:
The rebuttal effectively addresses several of the reviewer’s technical and presentation-related concerns, but issues regarding qualitative evidence remain unresolved. T

he authors improve the clarity of the paper by revising the explanation of the tokenization process and the interaction between the graph embedder and the tokenizer. The rewritten sections provide clearer mathematical descriptions, explicitly describe the role of the graph encoder as a structural prior, and clarify how joint semantics, topology, and motion features are fused. In addition, the restructuring of the paper and the relocation of figures improve readability and address the reviewer’s concern that the tokenization details were difficult to follow.

The reviewer’s request to ablate the importance of the graph embedder is also satisfactorily addressed. The authors provide new ablation experiments that remove graph-structure attention and joint-name semantics, showing clear degradation in accuracy. These results demonstrate that the graph embedder plays a crucial role in providing joint identity information within the tokenization model.

However, the concern that the motion transfer and generation directions are not explored thoroughly remains only partially addressed. Although the authors significantly expand the qualitative demonstrations and include a wider range of cross-morphology transfer examples, the resulting visualizations do not appear particularly strong. Without direct qualitative comparisons to existing methods, it is still difficult to assess whether the proposed approach meaningfully outperforms prior work. As a result, the qualitative evidence remains insufficient to judge the relative effectiveness of the method.

In summary, the rebuttal successfully resolves concerns related to the clarity of the tokenization design, paper structure, and the importance of the graph embedder through additional ablations. Nevertheless, the lack of comparative qualitative results for motion transfer and generation leaves a key concern outstanding, as the current demonstrations alone do not clearly establish the method’s advantage over existing approaches.

**Reviewer Scores:**

It seems all reviewers would maintain their original scores.

---

### Decision · Program_Chairs · 2026-01-26

Reject